# An altered cell-specific subcellular distribution of translesion synthesis DNA polymerase kappa (POLK) in aging mouse neurons

**Mofida Abdelmageed, Premkumar Palanisamy, Victoria Vernail, Yuval Silberman, Shilpi Paul, Anirban Paul***

Neuroscience and Experimental Therapeutics, Penn State Milton S. Hershey Medical Center, Hershey, United States

## eLife Assessment

This manuscript details **important** findings that DNA polymerase kappa shows age-related changes in subcellular localization within different cell types in the brains of mice, from the nucleus in young cells to the cytoplasm in old cells. The authors' findings suggest that age-related alterations in POLK localization could drive mechanistic and functional changes in the aging brain. The authors provide **solid** evidence for their study, with data broadly supporting their claims with minor weaknesses.

**\*For correspondence:**
amp7167@psu.edu

**Abstract** Genomic stability is critical for cellular function; however, in the central nervous system, highly metabolically active differentiated neurons are challenged to maintain their genome over the organismal lifespan without replication. DNA damage in neurons increases with chronological age and accelerates in neurodegenerative disorders, resulting in cellular and systemic dysregulation. Distinct DNA damage response strategies have evolved with a host of polymerases. The Y-family translesion synthesis (TLS) polymerases are well known for bypassing and repairing damaged DNA in dividing cells. However, their expression, dynamics, and role, if any, in enduring postmitotic differentiated neurons of the brain are completely unknown. We show through systematic longitudinal studies for the first time that DNA polymerase kappa (POLK), a member of the Y-family polymerases, is highly expressed in mouse neurons. With chronological age, there is a progressive and significant reduction of nuclear POLK with a concomitant accumulation in the cytoplasm that is predictive of brain tissue age. The reduction of nuclear POLK in old brains is congruent with an increase in DNA damage markers. The nuclear POLK colocalizes with damaged sites and DNA repair proteins. The cytoplasmic POLK accumulates with stress granules and endo/lysosomal markers. Nuclear POLK expression is significantly higher in GABAergic interneurons (INs) compared to excitatory pyramidal neurons and lowest in non-neurons, possibly reflective of the inherent biological differences such as firing rates and neuronal activity. INs associated with microglia have significantly higher levels of cytoplasmic POLK in old age. Finally, we show that neuronal activity itself can lead to an increase in nuclear POLK levels and a reduction of the cytoplasmic fraction. Our findings open a new avenue in understanding how different classes of postmitotic neurons deploy TLS polymerase(s) to maintain their genomic integrity over time, which will help design strategies for longevity, healthspan, and prevention of neurodegeneration.

## Introduction

Throughout the organismal lifespan, continuous exposure to exogenous and endogenous agents can damage one or both strands of the DNA thus causing a persistent threat to genomic integrity (*Iyama and Wilson, 2013*; *Yan and Vaziri, 2020*; *Chatterjee and Walker, 2017*; *Tubbs and Nussenzweig, 2017*). Due to their long lifespan, increased transcriptional activity, and large energy demands, neurons are highly susceptible to cumulative DNA damage (*Welch and Tsai, 2022*; *Caldecott et al., 2022*; *Wu et al., 2021*; *Reid et al., 2021*; *Madabhushi et al., 2014*). Neuronal genomic enhancer regions are hotspots for constant high levels of DNA single-strand breaks (SSBs) repair synthesis process, thus revealing that neurons are undergoing localized and continuous DNA breakage (*Reid et al., 2021*; *Wu et al., 2021*). In addition, the threat of genomic instability can also be contributed by normal 'programmed' DNA breaks like topoisomerase-induced breaks for the removal of topological stress during transcription. High levels of such programmed DNA breakage occur in neurons during development, differentiation, and maintenance (*Caldecott et al., 2022*). Recently, it was shown that programmed SSBs in the enhancer regions can be repaired by Poly(ADP-Ribose) Polymerase 1 (PARP1) and X-ray cross-complementing protein 1 (XRCC1)-mediated pathways (*Wu et al., 2021*). XRCC1 is a DNA repair scaffold protein that plays a role in multiple repair pathways including base excision repair (BER) and single-strand break repair (SSBR) (*London, 2020*). Due to high mitochondrial activity in the neurons, which consumes about 20% of the body's oxygen supply (*Attwell and Laughlin, 2001*), neurons exhibit high levels of reactive oxygen species (ROS)-mediated DNA damage. ROS can damage DNA by oxidizing bases and creating abasic sites (*Lindahl and Barnes, 2000*; *Madabhushi et al., 2014*; *Tubbs and Nussenzweig, 2017*). One of the most common ROS-induced DNA modifications is 8-oxo-7,8-dihydroguanine (8oxo-dG) (*Amente et al., 2019*; *Ding et al., 2017*), a lesion that is often resolved through BER (*Markkanen, 2017*). Unrepaired SSBs hinder transcription and can also lead to double-strand breaks (DSBs). A strong association exists between unrepaired DNA damage causing loss of genomic integrity with aging and neurodegeneration (*Caldecott et al., 2022*; *Welch and Tsai, 2022*). Hence, survival strategies have evolved to preserve genomic stability through multiple DNA damage response (DDR) pathways, allowing the neurons to function effectively. Impaired DDR leads to genomic instability, triggering signaling cascades, stalling transcription, and creating mutagenesis, thus altering the cellular fate toward anomalous cell cycle activation, apoptosis, or senescence, which are hallmarks of aging and age-associated diseases (*Hoeijmakers, 2009*; *Madabhushi et al., 2014*; *Chow and Herrup, 2015*; *McKinnon, 2017*; *Tubbs and Nussenzweig, 2017*; *Welch and Tsai, 2022*).

One unique mechanism that bypasses DNA adducts, synthesizes DNA past damage, and functions in post-replication DNA repair is through translesion synthesis (TLS) DNA polymerases (*Sale et al., 2012*; *Gao et al., 2017*; *Powers and Washington, 2018*; *Lehmann, 2006*; *Paniagua and Jacobs, 2023*). TLS polymerases are well-studied in dividing cells, and the Y-family TLS polymerases include Rev1, Pol kappa (Polk), Pol iota (Poli), and Pol eta (Polh) (*Guo et al., 2009*; *Sale et al., 2012*; *Vaisman and Woodgate, 2017*). However, their role in postmitotic cells is not explored. There is evidence of the role of POLK in dorsal root ganglion (DRG) neurons upon the treatment of genotoxic agents like cisplatin, where POLK is upregulated (*Zhuo et al., 2018*) and is essential for efficiently and accurately repairing cisplatin crosslinks (*Jha and Ling, 2018*). POLK can perform error-free TLS across bulky DNA adducts at the N2 position of guanine induced by the genotoxic agent benzo[a]pyrene-dihydrodiol-epoxide (BPDE) and mitomycin C (*Kanemaru et al., 2017*; *Ogi et al., 2002*; *Avkin et al., 2004*). POLK can also bypass other DNA lesions, including 8oxo-dG caused by oxidative stress in association with Werner syndrome helicase (WRN) protein (*Jałoszyński et al., 2005*; *Kamiya and Kurokawa, 2012*; *Haracska et al., 2002*; *Maddukuri et al., 2014*) and through strand break repair mechanism (*Zhang et al., 2013*). *Polk*$^{-/-}$ mice showed survival defects, spontaneous mutations, and sensitivity to BPDE (*Stancel et al., 2009*; *Singer et al., 2013*). Evidence supports that POLK can function in non-S phase cells through an NER-dependent mechanism to protect from UV-induced cytotoxic lesions (*Ogi and Lehmann, 2006*; *Sertic et al., 2018*). *Polk*$^{-/-}$ mouse embryo fibroblasts were sensitive to hydrogen peroxide and showed defects in both SSB and DSB repair, suggesting that Polk may have an important role in the oxidative stress-induced DNA repair process (*Zhang et al., 2013*). Although POLK is associated with multiple DNA repair mechanisms, it remains unknown if normative age-associated DNA damage will also recruit POLK and how it may function in postmitotic neurons.

Unlike the long-living postmitotic neurons that endure DNA damage, the non-neuronal (NN) cells can go through the cell cycle and are replaceable. Due to the non-replicative status of differentiated

neurons in mature circuits, it is critical to understand the different repair strategies and mechanisms that postmitotic neurons employ to maintain their genomic integrity, which will help design therapies for human longevity and the prevention of neurodegeneration. Here, we characterize the expression of POLK in the brain, its subcellular localization, how it is altered with chronological age, cell type-specific variability, and the microglial engagement with neurons, and neuronal activity in wild-type mouse brains undergoing healthy aging under unstressed conditions.

## Results

### A progressive age-associated shift in subcellular localization of POLK

POLK has been studied mostly in proliferating cells; however, its levels are found to be highest in the G0 phase of the cell cycle and upregulated upon exposure to the exogenous agent cisplatin in dorsal root neurons (*Zhuo et al., 2018*). Hence, we explored whether POLK is expressed in the brain, where both replicating and non-replicating cells make it a unique tissue environment. Since neurons endure an onslaught of a lifetime of DNA damage due to activity, genotoxic agents, and physiological age, we investigated if the levels of endogenous POLK in the mice brain also vary as a function of age. In addition, since POLK showed subcellular relocalization in cancer cells undergoing stress (*Temprine et al., 2020*; *Paul et al., 2023*), we further explored whether POLK in the brain shows similar changes during normative aging in C57BL/6J (RRID:IMSR_JAX:000664) wild-type mice.

We first validated using mouse primary cortical neurons and other murine cells a published POLK antibody (sc-166667) raised against human POLK protein targeting the epitope 131–310 amino acids, which has complete sequence homology between mouse and human POLK (*Figure 1—figure supplement 1A*). Knockdown of POLK in mouse primary cortical neurons (*Figure 1—figure supplement 1B*) showed a reduction in POLK levels using immunofluorescence (IF) (*Figure 1A1*), and western blot (WB) (*Figure 1A*, *Figure 1—figure supplement 1C*); a similar reduction of two POLK bands (~99 and 120 kDa) was observed in other efficiently transfectable murine cells like Neuro-2A (*Figure 1—figure supplement 1D*) and 4T1 (*Figure 1—figure supplement 1E*). We used another anti-POLK (A12052) that showed a similar POLK staining pattern as sc-166667 in mouse brain by IF (*Figure 1—figure supplement 1F*). From here on onwards for all assays we used anti-POLK antibody SC-166667. We observed a decrease in nuclear POLK bands by WB in mouse cortex unsorted cell nuclear lysate between 7 and 22 months; however, we did not see any changes in the cytoplasmic POLK (*Figure 1B*, *Figure 1—figure supplement 1G*). This prompted a longitudinal unbiased survey of POLK expression across multiple cortical areas at the subcellular level using IF to visualize the spatial–temporal changes of POLK with age in a healthy wild-type mouse brain; young 1 month, middle age 10 months, and early-old age 18 months mice (*Figure 1C*).

To visualize the changes at the subcellular level, we developed an automated image analysis pipeline using CellProfiler (*Stirling et al., 2021*) that segments the nucleus and cytoplasm using the fluorescent Nissl signal that primarily stains cell bodies in brain sections (*Paul et al., 2008*). POLK is detected as small 'speckles' inside the nucleus at a young age (1–2 months) and larger 'granules' can be seen in the cytoplasm at progressively older time points (>9 months) (*Figure 1D*). In young brains, both the normalized counts of nuclear POLK speckles and their size are consistent across the cingulate cortex (Cg1), motor cortex (M1 and M2), and somatosensory cortex (S1). A comparison of the nuclear POLK speckle counts and size using ANCOVA shows a significant age-associated decline with medium to large effect sizes across the four brain areas surveyed (*Figure 1E–G*, *Tables 1–3*) in agreement with the immunoblot assay. On the other hand, in the cytoplasm, there is a significant age-associated accumulation of POLK in the form of increasingly larger-sized granules with decreasing total counts (*Figure 1E, F, H*, *Tables 4–6*). Since cytoplasmic accumulation is non-homogeneous in the form of granules, suggesting cytoplasmic POLK accumulation potentially in cytoplasmic organelles or membrane-less biomolecular aggregates. Our cytoplasmic fractionation procedure did not fully extract proteins from such organelles or aggregates, which possibly explains the discrepancy between cytoplasmic POLK observed by biochemical and cell biological assays.

To check if the age-associated decline in nuclear POLK is a more generalized phenomenon in Y-family TLS polymerases, we tested the qualitative expression of REV1 and POL iota (POLI) in the mice brain. REV1 protein shows similar expression patterns as clusters of speckles inside the nucleus, a similar qualitative decline in nuclear expression, and an increase in cytoplasmic granules in M1 and

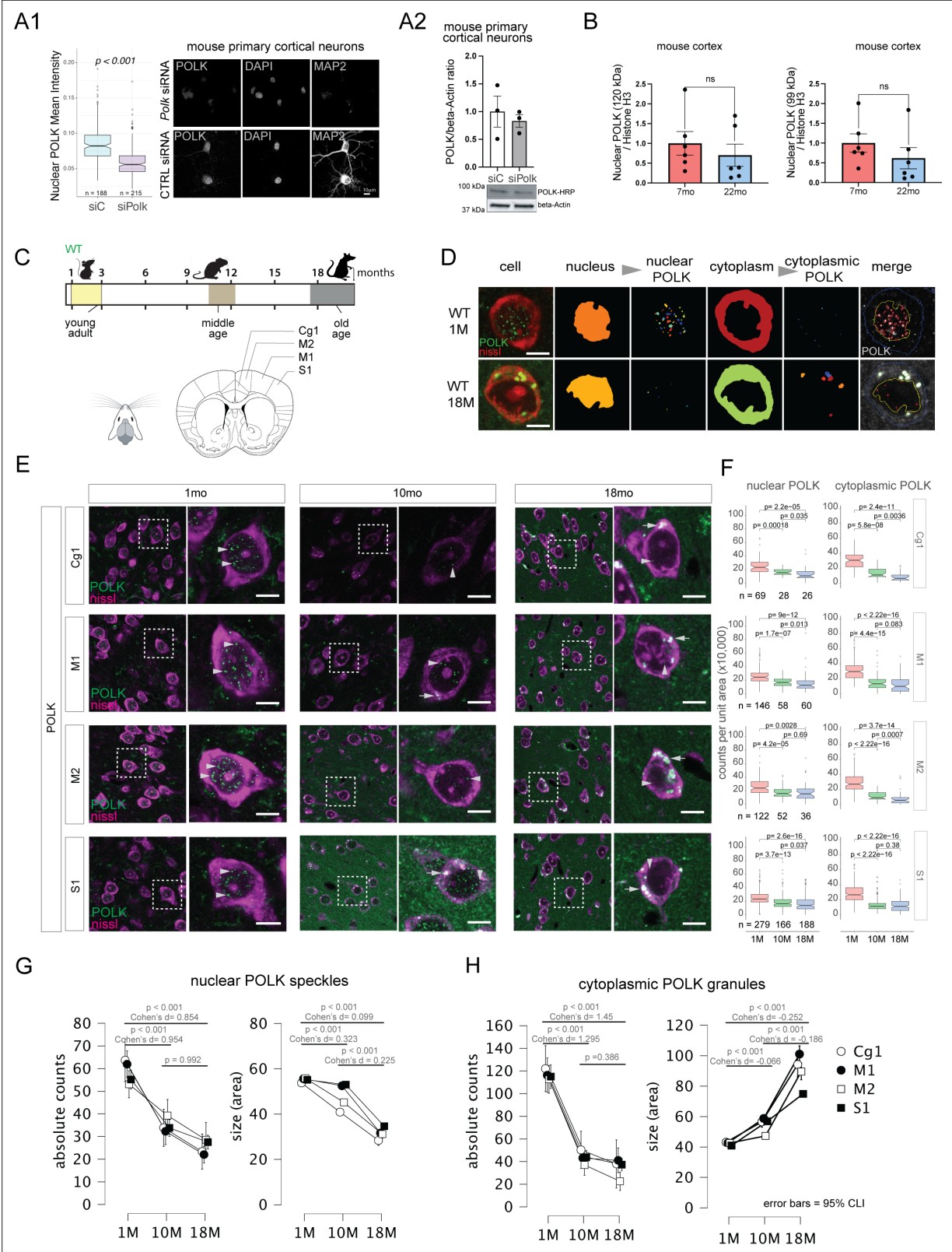

**Figure 1.** POLK subcellular expression changes with increasing age, across multiple cortical regions. (**A1**) Validation of anti-POLK antibody (sc-166667) by immunofluorescence on mouse primary cortical neuronal cultures treated with Polk siRNA (top row) showing a marked reduction in nuclear POLK levels after 48 hr, compared to scrambled siRNA control (bottom row). (**A2**) Western blot of mouse primary cortical neuronal culture treated with siPOLK whole cell lysate immunostained with sc-166667 HRP anti-POLK (HRP conjugated) antibody showing ~20% reduction in POLK protein levels compared

*Figure 1 continued on next page*

*Figure 1 continued*

to siControl. (**B**) Quantitation of 120 and 99 kDa POLK bands from western blot using sc-166667 HRP antibody shows a decreasing trend of nuclear POLK levels with progressive age from unsorted cells of the whole mouse cortex. Full blot shown in *Figure 1—figure supplement 1C*. (**C**) Experimental design to longitudinally compare POLK cellular localization in mice brains. (**D**) Immunofluorescence (IF) followed by imaging and analysis using Cell Profiler of mouse brain sections. Visualization of POLK speckles (green) expression using immunohistochemistry in cells labeled by fluorescent-nissl (red) from wild-type mouse brain aged 1 and 18 months. Subcellular compartments were segmented and POLK was detected and measured separately inside the nucleus and in the cytoplasm. (**E**) Representative low (×20) and high magnification (×63) images of Polk expression using IF from Cg1, M1, M2, and S1 cortical regions in ages 1 month (N = 3), 10 months (N = 2), and 18 months (N = 3). POLK (green) and Nissl depicting all cells (purple) (scale bar = 10 μm). (**F**) Boxplot showing nuclear and cytoplasmic POLK counts per unit area for each brain region in (C) grouped by age. *n* denotes the numbers of cells measured per time points under each boxplot. (**G**) Means plots with 95% confidence intervals showing the nuclear POLK speckle count and size decreasing with increasing age in Cg1, M1, M2, and S1 cortical areas. (**H**) Means plots with 95% confidence intervals show the cytoplasmic POLK granule count decreasing with a concomitant increase in size with age in Cg1, M1, M2, and S1 cortical areas.

The online version of this article includes the following source data and figure supplement(s) for figure 1:

**Source data 1.** Annotated original blots corresponding to *Figure 1*, panel A2.

**Source data 2.** Raw scans of original blots to *Figure 1*, panel A2.

**Source data 3.** CSV files, Prism files and R script related to *Figure 1*.

**Figure supplement 1.** Supplement to POLK subcellular expression changes with increasing age, across multiple cortical regions.

**Figure supplement 1—source data 1.** Annotated original blots corresponding to *Figure 1—figure supplement 1*.

**Figure supplement 1—source data 2.** Raw scans of original blots to *Figure 1—figure supplement 1*.

S1 brain areas at 1, 10, and 18 months. Comparatively, POLI nuclear expression showed overall lower levels, with decreased age-associated alterations in nuclear expression; however, it still showed age-associated cytoplasmic accumulation (*Figure 1—figure supplement 1H*). Antibody against Pol eta (POLH) showed a very weak signal or failed.

## Nuclear POLK colocalizes with DDR and repair proteins

Various pathways in the DDR have evolved to detect and repair different types of lesions that threaten the genome. Among these, ROS pose a significant threat to the neurons (*Attwell and Laughlin, 2001*), creating ROS-induced DNA modifications 8oxo-dG (*Lindahl and Barnes, 2000*; *Madabhushi et al., 2014*; *Tubbs and Nussenzweig, 2017*). Repair of 8oxo-dG lesions primarily occurs through BER, where a glycosylase enzyme identifies and removes the damaged base, followed by cleavage of the DNA backbone by apurinic/apyrimidinic endonuclease 1 (APE1), resulting in an SSB. This inter-mediate can then be processed by polymerase beta (POLb), which fills in the gap with the correct nucleotide, and the nick is sealed by ligase III (LIG3). Along with SSBs, DSBs are more harmful due to their potential for toxicity. Despite this toxicity, DSBs also play crucial roles in cellular physiology. For instance, neurons require DSBs to facilitate the expression of immediate early genes (*Suberbielle et al., 2013*; *Madabhushi et al., 2015*; *Alt and Schwer, 2018*), particularly through transcriptional activity in postmitotic neurons. Additionally, SSBs may also lead to DSBs (*Cannan and Pederson, 2016*). DSB repair in postmitotic cells is predominantly managed by the non-homologous end joining (NHEJ) pathway. In canonical NHEJ, DSBs are initially recognized and bound at their ends by KU70/80 and DNA-dependent protein kinase (PRKDC). Subsequently, they are directly reconnected through ligation mediated by Ligase IV (LIG4), X-Ray Repair Cross Complementing 4 (XRCC4), and XRCC4-like

**Table 1.** Absolute counts of nuclear Polk speckles.

**ANCOVA post hoc comparisons – age, with nuclear area as covariate**

|  |  | Mean Diff | SE | t | Cohen's d | $p_{tukey}$ |  | $p_{bonf}$ |  |
|---|---|---|---|---|---|---|---|---|---|
| 1M | 10M | 18.432 | 1.806 | 10.203 | 0.854 | <0.001 | *** | <0.001 | *** |
| 1M | 18M | 20.584 | 1.918 | 10.733 | 0.954 | <0.001 | *** | <0.001 | *** |
| 10M | 18M | 2.152 | 2.212 | 0.973 | 0.1 | 0.594 |  | 0.992 |  |

Note. Results are averaged over the levels of brain area.

Note. p-value adjusted for comparing a family of 3.

*p < 0.05, **p < 0.01, ***p < 0.001.

**Table 2.** Nuclear POLK counts per unit area × age × brain area.

**ANCOVA post hoc comparisons – age × brain area, with nuclear area as covariate**

| | | Mean Diff | SE | t | Cohen's d | p_tukey | | p_bonf | |
|---|---|---|---|---|---|---|---|---|---|
| 1M Cg1 | 10M Cg1 | 22.276 | 4.841 | 4.602 | 1.032 | <0.001 | *** | <0.001 | *** |
| 1M Cg1 | 18M Cg1 | 25.265 | 4.987 | 5.066 | 1.171 | <0.001 | *** | <0.001 | *** |
| 10M Cg1 | 18M Cg1 | 2.99 | 5.881 | 0.508 | 0.139 | 1 | | 1 | |
| 1M M1 | 10M M1 | 18.729 | 3.366 | 5.564 | 0.868 | <0.001 | *** | <0.001 | *** |
| 1M M1 | 18M M1 | 24.795 | 3.341 | 7.421 | 1.149 | <0.001 | *** | <0.001 | *** |
| 10M M1 | 18M M1 | 6.066 | 3.976 | 1.526 | 0.281 | 0.933 | | 1 | |
| 1M M2 | 10M M2 | 17.852 | 3.576 | 4.992 | 0.827 | <0.001 | *** | <0.001 | *** |
| 1M M2 | 18M M2 | 14.258 | 4.105 | 3.474 | 0.661 | 0.026 | * | 0.035 | * |
| 10M M2 | 18M M2 | −3.593 | 4.699 | −0.765 | −0.167 | 1 | | 1 | |
| 1M S1 | 10M S1 | 14.873 | 2.117 | 7.027 | 0.689 | <0.001 | *** | <0.001 | *** |
| 1M S1 | 18M S1 | 18.018 | 2.057 | 8.758 | 0.835 | <0.001 | *** | <0.001 | *** |
| 10M S1 | 18M S1 | 3.144 | 2.293 | 1.371 | 0.146 | 0.969 | | 1 | |

Note. p-value adjusted for comparing a family of 12; only relevant comparisons are shown *p < 0.05, **p < 0.01, ***p < 0.001.

factor (XLF). At DSB sites, the protein kinase ATM phosphorylates numerous downstream substrates such as histone variant H2A.X that form gH2AX foci, which are docking sites for DNA repair proteins like p53-binding protein 1 (53BP1) that promote NHEJ-mediated DSB repair (*Mirman and de Lange, 2020*; *Panier and Boulton, 2014*).

There are hints that TLS polymerases play a role in multiple repair pathways in dividing cells; for example, POLK inserts the correct base opposite the 8oxo-dG lesion and extends from dC:8oxo-dG base pairs, supporting a potential role in BER (*Maddukuri et al., 2014*). We previously identified DNA repair pathway proteins by performing iPoKD-MS (isolation of proteins on Pol kappa synthesized DNA followed by mass spectrometry), where proteins were captured and bound to the nascent DNA synthesized by Polk in human cell lines (*Paul et al., 2023*). Hence, we tested whether neuronal nuclear POLK is associated with BER and NHEJ pathway proteins identified in our iPoKD-MS datasets. We tested five NHEJ pathway markers and four BER pathway markers at middle age (9–12 months) in the wild-type mouse brain cortex and analyzed the colocalization of these proteins and the markers at the nuclear POLK sites (*Figure 2A*). Images from the mouse cortex with at least three biological replicates per age group were parsed into excitatory pyramidal neurons (PNs) and inhibitory interneurons (INs). Nissl+, NeuN+, and Gad67+ cells were labeled as IN class and Nissl+, NeuN+, and Gad67− as PN class neurons. Each IN and PN class cell body was subcellularly segmented into nucleus and cytoplasm, and individual nuclear POLK speckles were identified. NHEJ and BER marker intensities were then measured within each nuclear POLK speckle (*Figure 2B*).

**Table 3.** Size (area) of nuclear Polk speckles.

**ANOVA post hoc comparisons – age**

| | | Mean difference | SE | t | Cohen's d | p_tukey | | p_bonf | |
|---|---|---|---|---|---|---|---|---|---|
| 1M | 10M | 7.195 | 0.431 | 16.708 | 0.099 | <0.001 | *** | <0.001 | *** |
| 1M | 18M | 23.617 | 0.578 | 40.861 | 0.323 | <0.001 | *** | <0.001 | *** |
| 10M | 18M | 16.422 | 0.653 | 25.143 | 0.225 | <0.001 | *** | <0.001 | *** |

Note. Results are averaged over the levels of brain area.
Note. p-value adjusted for comparing a family of 3.
*p < 0.05, **p < 0.01, ***p < 0.001.

**Table 4.** Absolute counts of cytoplasmic Polk granules.

**Post hoc comparisons – age, with cytoplasmic area as covariate**

|  |  | Mean Diff | SE | t | Cohen's d | $p_{tukey}$ |  | $p_{bonf}$ |  |
|---|---|---|---|---|---|---|---|---|---|
| 1M | 10M | 59.09 | 3.824 | 15.451 | 1.295 | <0.001 | *** | <0.001 | *** |
| 1M | 18M | 66.164 | 3.997 | 16.553 | 1.45 | <0.001 | *** | <0.001 | *** |
| 10M | 18M | 7.074 | 4.654 | 1.52 | 0.155 | 0.282 |  | 0.386 |  |

Note. Results are averaged over the levels of brain area.
Note. p-value adjusted for comparing a family of 3.
*p < 0.05, **p < 0.01, ***p < 0.001.

To quantify the association between the DNA damage marker intensities (gH2AX, 53BP1, and 8-oxodG) and POLK levels across age groups and cell types, we first computed Pearson correlation coefficients (*r*) at the single-cell level. However, because thousands of cells originate from biological replicates, we modeled the relationship using generalized estimating equations (GEE), which estimate population-level effects while explicitly accounting for intra-cluster dependence. GEE assumes a marginal (population-average) linear relationship between the variables and uses a working correlation structure to capture the fact that all cells from the same replicate share similar covariance. For each age group × cell type panel, the p-value reported reflects a hypothesis test of the slope relating POLK intensity to the nuclear markers tested within the GEE framework, that is, whether the population-level association is significantly different after accounting for replicate-level clustering. These p-values were subsequently Bonferroni-adjusted across panels to control the family-wise error rate. Collectively, this approach provides a conservative estimate of statistical significance and yields inference that is valid at the population level and should be interpreted as population-average relationships across biological replicates, capturing broad trends in how marker intensities covary with POLK across different ages and cell types.

In middle age, for both IN and PN classes, we observed that POLK intensity positively correlated with DSB marker gH2AX (*Figure 2C1, C2*), NHEJ markers 53BP1 (*Figure 2D1, D2*) and PRKDC (*Figure 2F1*), and ROS-mediated damage marker 8oxo-dG (*Figure 2E1, E2*). Interestingly, BER pathway marker APE1 and, to a lesser degree, LIG3 were only present in POLK speckles in INs but not in PNs (*Figures 2F2, F3*). In the NHEJ pathway, only PRKDC showed high colocalization with nuclear

**Table 5.** Absolute counts of cytoplasmic Polk granule comparison by age and brain area.

**ANOVA post hoc comparisons – age × brain area, with cytoplasmic area as covariate**

|  |  | Mean Diff | SE | t | Cohen's d | $p_{tukey}$ |  | $p_{bonf}$ |  |
|---|---|---|---|---|---|---|---|---|---|
| 1M Cg1 | 10M Cg1 | 57.944 | 10.232 | 5.663 | 1.27 | <0.001 | *** | <0.001 | *** |
| 1M Cg1 | 18M Cg1 | 80.174 | 10.502 | 7.634 | 1.757 | <0.001 | *** | <0.001 | *** |
| 10M Cg1 | 18M Cg1 | 22.23 | 12.432 | 1.788 | 0.487 | 0.824 |  | 1 |  |
| 1M M1 | 10M M1 | 59.109 | 7.094 | 8.333 | 1.295 | <0.001 | *** | <0.001 | *** |
| 1M M1 | 18M M1 | 60.769 | 7.009 | 8.67 | 1.332 | <0.001 | *** | <0.001 | *** |
| 10M M1 | 18M M1 | 1.661 | 8.403 | 0.198 | 0.036 | 1 |  | 1 |  |
| 1M M2 | 10M M2 | 58.471 | 7.572 | 7.722 | 1.281 | <0.001 | *** | <0.001 | *** |
| 1M M2 | 18M M2 | 66.588 | 8.679 | 7.672 | 1.459 | <0.001 | *** | <0.001 | *** |
| 10M M2 | 18M M2 | 8.117 | 9.896 | 0.82 | 0.178 | 1 |  | 1 |  |
| 1M S1 | 10M S1 | 60.836 | 4.466 | 13.623 | 1.333 | <0.001 | *** | <0.001 | *** |
| 1M S1 | 18M S1 | 57.122 | 4.343 | 13.153 | 1.252 | <0.001 | *** | <0.001 | *** |
| 10M S1 | 18M S1 | –3.713 | 4.853 | –0.765 | –0.081 | 1 |  | 1 |  |

Note. p-value adjusted for comparing a family of 12; only relevant comparisons are shown *p < 0.05, **p < 0.01, ***p < 0.001.

**Table 6.** Size (area) of cytoplasmic Polk granules.

**Post hoc comparisons – age**

| | | Mean Diff | SE | t | Cohen's d | p_tukey | | p_bonf | |
|---|---|---|---|---|---|---|---|---|---|
| 1M | 10M | −12.531 | 0.667 | −18.778 | −0.066 | <0.001 | *** | <0.001 | *** |
| 1M | 18M | −47.599 | 0.904 | −52.653 | −0.252 | <0.001 | *** | <0.001 | *** |
| 10M | 18M | −35.068 | 1.036 | −33.834 | −0.186 | <0.001 | *** | <0.001 | *** |

Note. Results are averaged over the levels of area.
Note. p-value adjusted for comparing a family of 3. ***p < 0.001.

POLK in both IN and PN, but other NHEJ proteins like KU70 and XRCC4 did not colocalize, and their intensities were not correlated with POLK levels (*Figure 2—figure supplement 1*).

Since both gH2AX and 8oxo-dG were positively associated with the nuclear POLK speckles, we wanted to examine if their colocalization and intensity correlation with POLK levels varies with progressive age. So, we further tested early-old (18 months) and late-old (24–27 months) time points. This showed that the temporal profile of gH2AX and 8oxo-dG association with POLK are distinct. Broadly, peak colocalization of gH2AX in POLK speckles was at the early-old time point (*Figure 2C3*), whereas the presence of 8oxo-dG in POLK speckles declines in early-old age but strikingly increases in the late-old age group (*Figure 2E3*). The temporal profile of gH2AX intensities in POLK speckles and gH2AX-POLK correlation was mirrored by 53BP1, which is well-known to promote DSB repair through the NHEJ pathway, suggesting higher levels of DSBs at early-old age and spatio-temporal association of POLK. To further inspect if there is differential DNA damage between IN and PN, for each age group, we modeled the mean difference in marker intensities for gH2AX, 53BP1, and 8oxo between IN and PN using the same GEE approach. The primary p-values test whether the cluster-adjusted IN versus PN means are significantly different. Across age groups, p-values were corrected using the Benjamini–Hochberg method to control the false discovery rate. Though both INs and PNs showed similar profiles, nuclear POLK speckles had consistently higher levels of gH2AX intensities in INs compared to PNs with the large effect size for 53BP1 throughout all three age groups studied (*Figure 2D3*), suggesting that nuclear POLK plays a role at DSB sites in INs more compared to PNs. In contrast, there was no difference to ROS-induced 8oxo-dG intensities between IN and PNs, as well as decreased 8oxo-dG intensities in middle age (*Figure 2E3*). We observed POLK association with BER proteins in middle age and reduction in ROS-induced DNA damage, suggesting that POLK plays a role in the BER pathway during middle age mostly in INs compared to PNs, and perhaps this capacity of POLK is lost with aging.

Interestingly, we also observed with an increase in age, an elevated neuronal cytoplasmic gH2AX accumulation in the form of granules that colocalized with cytoplasmic POLK. During normal and premature aging syndromes, accumulation of gH2AX-enriched cytoplasmic DNA as cytoplasmic chromatin fragments (CCFs) were reported that correlate with inflammatory gene expression and activation of cGAS-STING pathways (*Miller et al., 2021a*; *Lan et al., 2019*; *Miller et al., 2021b*). In senescent cells, CCFs bud off the nucleus and are processed via lysosomes/autophagy-mediated proteolytic processing (*Ivanov et al., 2013*), which often accumulates due to lysosomal defects that impair proteolytic clearance (*Carmona-Gutierrez et al., 2016*). Our observation of cytoplasmic POLK granules with gH2AX hints toward cytoplasmic sequestration potentially impairing its DNA repair activity and accumulation in lysosomes.

## POLK in the cytoplasm is associated with stress granules and lysosomes in old brains

During the process of aging, dysregulated protein homeostasis can occur through widespread intra-cellular protein aggregates (*Taylor and Dillin, 2011*). A decline in the protein quality control system has been shown to impact the assembly and dynamic maintenance of stress granules (SGs), leading to increased cellular aggregation. SGs are nonmembrane assemblies of untranslated mRNAs, RNA-binding proteins, protein translation factors, and many non-RNA-binding proteins (*Cao et al., 2020*; *Guzikowski et al., 2019*), which can be induced by oxidative stress and other stressors (*Federico*

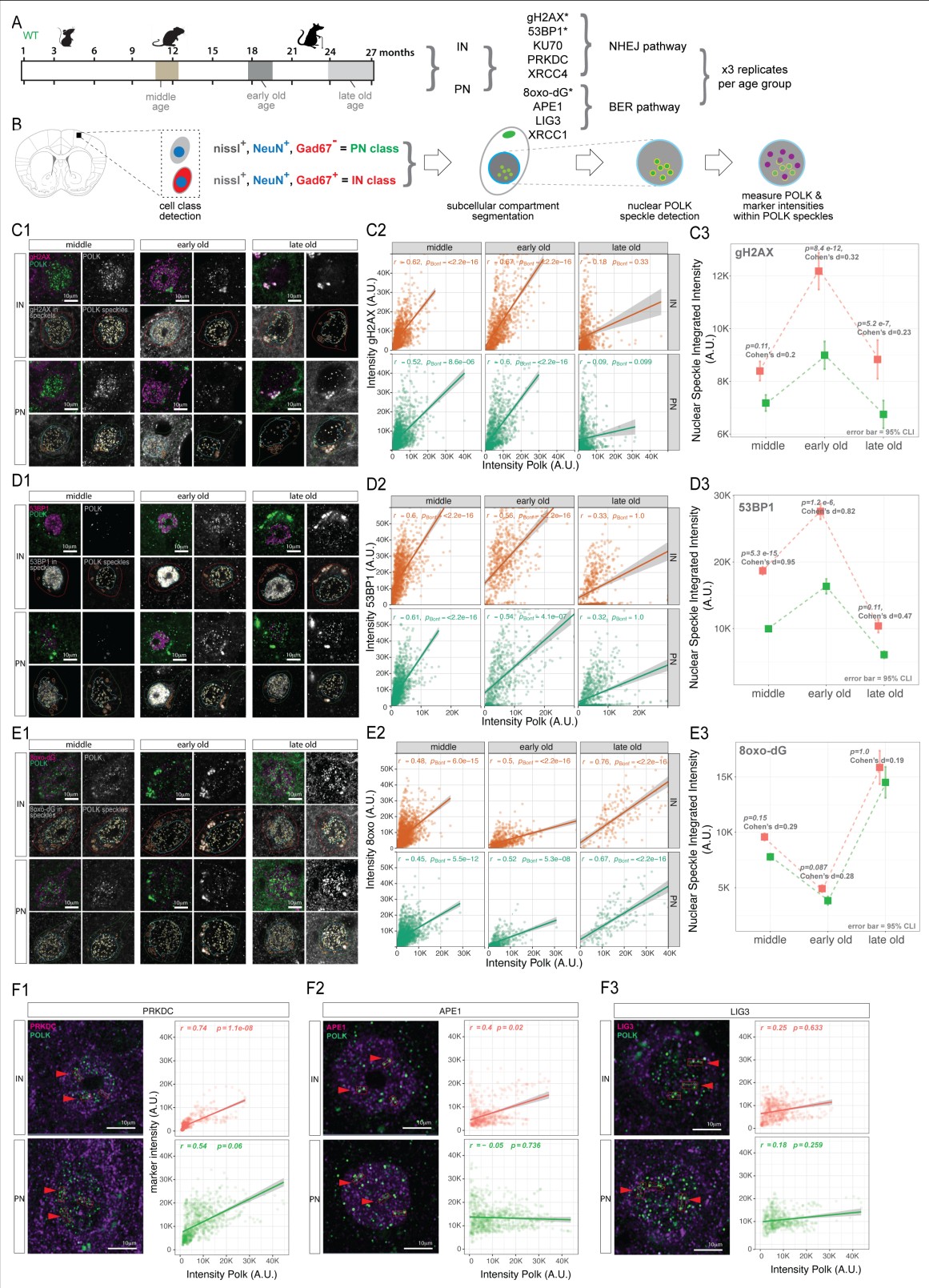

**Figure 2.** Colocalization and coexpression of POLK with DNA damage marker proteins in POLK nuclear speckles. (**A**) Experimental design to longitudinally compare POLK colocalization with various DNA damage markers in three age group mice brains. Asterisks denote markers that were studied in three age groups; the rest were evaluated in middle age only. (**B**) Schematic of cell class gating logic, registration, subcellular segmentation, POLK speckle detection, measurement of POLK counts, and POLK and DNA damage markers/repair protein intensities. (**C1**) Representative images

*Figure 2 continued on next page*

*Figure 2 continued*

showing colocalization of gH2AX and channel separation of POLK, gH2AX, and POLK speckle overlay on POLK and gH2AX. POLK speckles detected inside the nucleus are outlined in yellow. IN class cells are outlined in red, PN class in green. (**C2**) Scatterplots of gH2AX intensity measured inside POLK nuclear speckle (*y*-axis) were plotted against POLK intensities (*x*-axis) shown for INs (red) and PNs (green). The correlation coefficient and p-values are indicated for each. (**C3**) Dot mean plots with 95% confidence error bars of gH2AX intensities in the *y*-axis in INs (red) and PNs (green) plotted as a function of age (*x*-axis). (**D1**) Representative images showing colocalization of 53BP1 and channel separation of POLK, 53BP1, and POLK speckle overlay on POLK and 53BP1. POLK speckles detected inside the nucleus are outlined in yellow. IN class cells are outlined in red, and PN class in green. (**D2**) Scatterplots of 53BP1 intensity measured inside POLK nuclear speckle (*y*-axis) were plotted against POLK intensities (*x*-axis) shown for INs (red) and PNs (green). The correlation coefficient and p-values are indicated for each. (**D3**) Dot mean plots with 95% confidence error bars of 53BP1 intensities in the *y*-axis in INs (red) and PNs (green) plotted as a function of age (*x*-axis). (**E1**) Representative images showing colocalization of 8oxo-dG and channel separation of POLK, 8oxo-dG, and POLK speckle overlay on POLK and 8oxo-dG. POLK speckles detected inside the nucleus are outlined in yellow. IN class cells are outlined in red, and PN class in green. (**E2**) Scatterplots of 8oxo-dG intensity measured inside POLK nuclear speckle (*y*-axis) were plotted against POLK intensities (*x*-axis) shown for INs (red) and PNs (green). The correlation coefficients and p-values are indicated for each. (**E3**) Dot mean plots with 95% confidence error bars of 8oxo-dG intensities in the *y*-axis in INs (red) and PNs (green) plotted as a function of age (*x*-axis). Left column, merged 63X representative images of IN and PN, with red arrowheads showing line scan area where intensities of POLK and PRKDC (**F1**), POLK and APE1 (**F2**), and POLK and LIG3 (**F3**) were measured. The right columns in F1–F3 show corresponding IN- and PN-derived scatterplots where the *y*-axis shows PRKDC intensities (**F1**), APE1 (**F2**), LIG3 (**F3**), plotted against POLK intensities (*x*-axis). Separate plots are shown for INs (red) and PNs (green). The correlation coefficients and p-values are indicated for each.

The online version of this article includes the following source data and figure supplement(s) for figure 2:

**Source data 1.** CSV files, R and Python script related to *Figure 2*.

**Figure supplement 1.** Scatter plots showing intensity–intensity plots from INs (red) and PNs (green), where *y*-axis is KU70, XRCC1, and XRCC4, intensity levels plotted against POLK intensities in the *x*-axis.

*et al., 2012*; *Gandhi and Abramov, 2012*; *Patten et al., 2010*). Since we observed with increasing age elevated DNA damage markers (*Figure 2*), and POLK increasingly accumulating in the cytoplasm as 'granules' (*Figure 1*), we hypothesized that cytoplasmic POLK may be present with SGs. Ras-GTPase-activating protein (GAP)-binding protein 1 (G3BP1) is a key player in SG assembly often referred to as SG nucleator (*Sidibé et al., 2021*; *Aulas et al., 2015*; *Sahoo et al., 2018*), hence we tested if cytoplasmic POLK granules colocalize with G3BP1. In young (1 month) brains, there was minimal expression of G3BP1; however, by the early-old stage (18 months), cytoplasmic POLK is almost entirely colocalized with both G3BP1 and LAMP1 (*Figure 3B*, *Figure 3—figure supplement 1A, B*).

A decline in organelle homeostasis including the endosome-lysosome leading to compromised proteolytic capability is well established in aging neurons (*Cheng et al., 2018a*; *Nixon, 2020*). So, we tested colocalization of cytoplasmic POLK granules with established endo/lysosomal marker proteins. We observed partial colocalization with EEA1, an early endosomal marker (*Figure 3C*), suggesting cytoplasmic POLK in highly dynamic early endosomes that undergo continual fusion and fission with lysosomes to sort cargo (*Christoforidis et al., 1999*; *Sweet, 1999*). Lysosomes act as terminal degradative hubs for autophagic and endocytic substrates, supporting neuronal homeostasis necessary for growth, survival, and synaptic remodeling. Lysosomes exist as a continuum of heterogeneous intermediate organelles differing in morphology, membrane composition, hydrolase content, luminal pH, and function (*Cai et al., 2010*; *Farías et al., 2017*; *Padamsey et al., 2017*; *Klionsky and Emr, 2000*; *Levine and Klionsky, 2004*; *Maday et al., 2012*; *Maday and Holzbaur, 2014*; *Nixon, 2013*; *Saftig and Klumperman, 2009*). In neurons, lysosome-associated membrane proteins (LAMP1)-labeled organelles are heterogeneous, comprising intermediates of endocytic, autophagic pathways and in lysosomal biogenesis. We observed high colocalization of cytoplasmic POLK with LAMP1 in 18-month-old mice neurons (*Figure 3B*). However, not all LAMP1-positive compartments are degradative due to limited (or undetectable) hydrolase content (*Hollenbeck, 1993*; *Lee et al., 2011*; *Maday et al., 2012*; *Maday and Holzbaur, 2014*; *Cheng et al., 2015*; *Gowrishankar et al., 2015*; *Goo et al., 2017*; *Cheng et al., 2018b*; Cheng et al., 2018a). A hallmark of degradative lysosomes is the presence of active hydrolases, such as Cathepsins B and D and the glucocerebrosidase GCase. In our imaging, POLK partially colocalized with Cathepsin B (*Figure 3D*), colocalized robustly with Cathepsin D (*Figure 3E*), an aspartyl protease residing in the lumen of mature (late endosome) lysosomes often autolysosomes. However, there was minimal colocalization with GBA1/GCase (*Figure 3F*), a sphingolipid hydrolase that is also imported into mitochondria to support complex-I integrity (*Grabowski et al., 1990*; *Baden et al., 2023*; *Cheng et al., 2018b*). We however did not test any mitochondrial

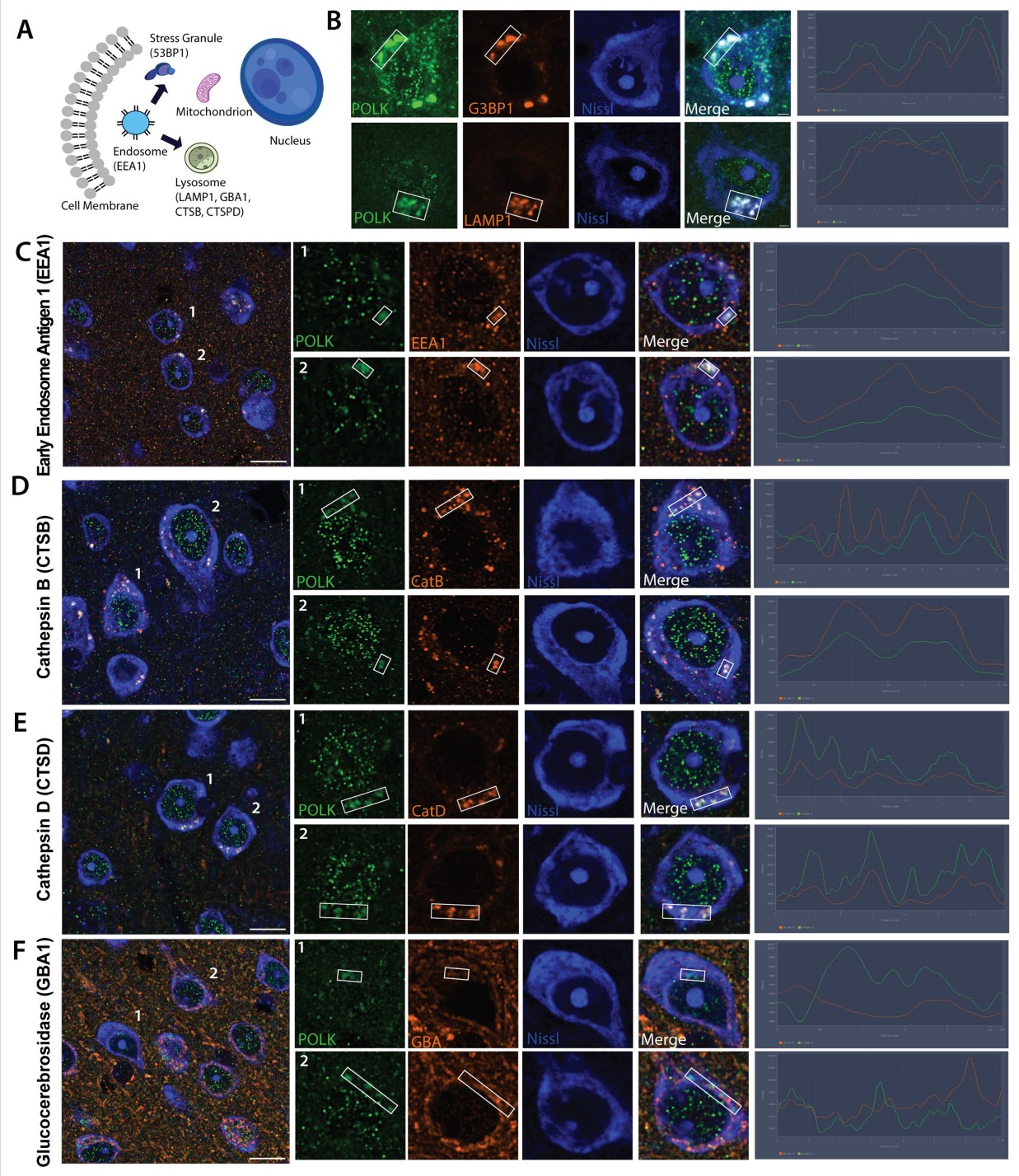

**Figure 3.** Cytoplasmic POLK expression colocalizing with stress granules and lysosomal proteins. (**A**) Schematic of cytoplasmic compartments and condensates and respective markers in parentheses that were assayed to colocalize with cytoplasmic POLK. (**B**) Cytoplasmic POLK (green) expression colocalizing with G3BP1 and LAMP1 (red) in fluorescent-nissl stained cells (blue) of mouse brain tissue in 18-month-old brain. Line scan in the boxed region shows POLK colocalizing with LAMP1 and G3BP1. (**C–F**) Cytoplasmic POLK (green), EEA1, CTSB, CTSD, and GBA1 (red) in fluorescent-nissl stained cells (blue) of mouse brain tissue in 18-month-old brain. Line scan in the boxed region shows level of POLK colocalization with the proteins. Highest colocalization with CTSD, partial with EEA1 and CTSB, and minimal GBA1.

The online version of this article includes the following figure supplement(s) for figure 3:

**Figure supplement 1.** Supplement to cytoplasmic POLK expression colocalizing with stress granules and lysosomal proteins.

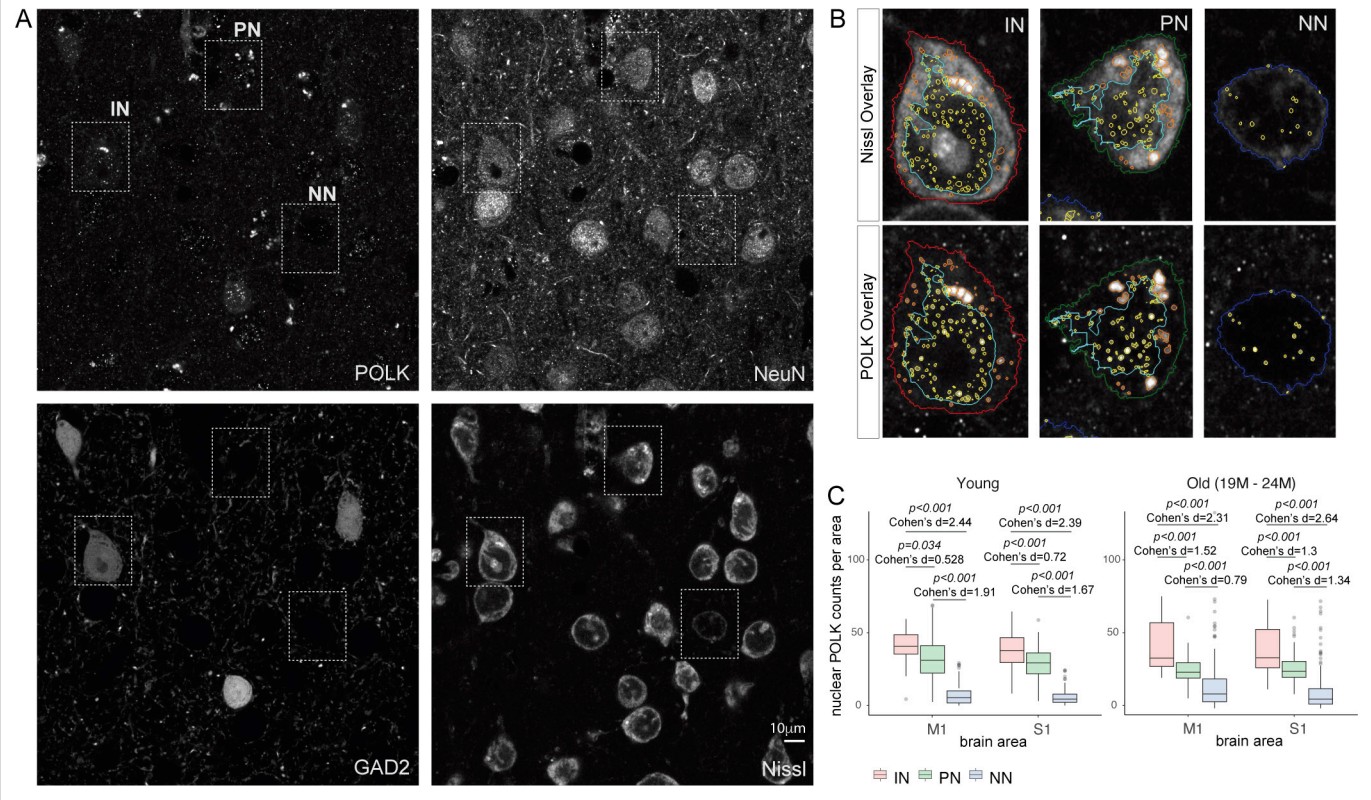

**Figure 4.** Nuclear POLK is differentially expressed based on cell type in old brains. (**A**) Representative image of the detection of inhibitory interneurons (INs), excitatory pyramidal neurons (PNs), and non-neuronal (NN) cell bodies using automated image analysis pipeline from a four-channel image of cortical areas from 19-month-old brain. (**B**) Magnified view of the dotted box from subpanel A showing overlay of the nuclear, cytoplasmic segmentation outlines and detection of nuclear POLK speckles and cytoplasmic POLK granules in IN, PN, and NN cells from wild-type mice cortical areas M1 and S1. (**C**) Boxplots from brain areas show the nuclear POLK count is higher in INs than PNs and NNs across cortical areas in both young and old age groups.

The online version of this article includes the following source data for figure 4:

**Source data 1.** CSV file and R Script of differential nuclear POLK expression by cell-types.

markers. Overall, the data support a model in which cytoplasmic POLK resides highly in endocytic/degradative compartments (CTSD-rich) and associates with SGs in aging neurons.

## Differentially altered POLK subcellular expression among excitatory, inhibitory, and NN cells in the cortex

To further test if the subcellular shift in POLK impacts all cells uniformly, we employed homozygous Gad2; Ai14 reporter to identify inhibitory INs from excitatory PNs and NN cells of young (1 month) and old (19–24 months) brains. Brain sections were co-immunostained with (1) anti-NeuN to detect pan-neuronal protein RBFOX3, (2) tdTomato to detect GABAergic neurons, (3) fluorescent Nissl to label all cells, and (4) anti-POLK to measure subcellular POLK levels. Our original image quantitation pipeline was modified to register cells positive for Gad2, NeuN, and Nissl as GABAergic INs, NeuN and Nissl positives as glutamatergic excitatory PNs, and Nissl positives only as NN cells (*Figure 4A*). The INs and PNs were further segmented into their nuclear and cytoplasmic compartments whereas NN subcellular segmentation was not implemented as their nucleus occupies most of the cellular morphology (*García-Cabezas et al., 2016*; *Figure 4B*). Hence in NN, the POLK signal was considered to primarily represent the nuclear compartment. We restricted our survey to the GABAergic inhibitory and glutamatergic excitatory neurons in cortical areas M1 and S1, which have extremely low (<0.01%) cholinergic and almost no dopaminergic, serotonergic, glycinergic, and histaminergic neuronal cell bodies (*Zhang et al., 2023*). Strikingly, at all ages across the cortical areas observed, the nuclear POLK signal was consistently highest in the cell group identified as INs, followed by PNs and NNs with large effect sizes (*Figure 4C*, *Tables 7 and 8*).

**Table 7.** Cell class nuclear Polk counts per age.

**Post hoc comparisons – age_bracket × cell type**

| | | Mean Diff | SE | t | Cohen's d | p_tukey | |
|---|---|---|---|---|---|---|---|
| Young IN | Young PN | 7.593 | 1.276 | 5.948 | 0.624 | <0.001 | *** |
| Young IN | Young NN | 29.403 | 1.266 | 23.231 | 2.418 | <0.001 | *** |
| Young PN | Young NN | 21.81 | 0.721 | 30.252 | 1.793 | <0.001 | *** |
| Early old IN | Early old PN | 17.191 | 1.391 | 12.362 | 1.414 | <0.001 | *** |
| Early old IN | Early old NN | 30.189 | 1.352 | 22.333 | 2.482 | <0.001 | *** |
| Early old PN | Early old NN | 12.998 | 0.762 | 17.066 | 1.069 | <0.001 | *** |

Note. Results are averaged over the levels of area.
Note. p-value adjusted for comparing a family of 6, only relevant comparisons shown. ***p < 0.001.

## Microglia associated with IN and PN have significantly higher levels of cytoplasmic POLK

Synapses and cell bodies are removed by brain microglia, and activated microglia have been associated with various neurological disorders. To test if neurons with higher cytoplasmic POLK granules are vulnerable to microglia engagement, we surveyed by immunostaining with microglia marker protein IBA1 in middle (11 months), early-old (18 months), and late-old (24–27 months) time points. Images were analyzed to register IBA1 + cells as microglia (MG class), GAD67+, NeuN+, and Nissl+ cells as IN class and NeuN+, Gad67−, and Nissl+ cells as PN class (*Figure 5A*). IN or PNs where MG cells are overlapping or in contact are scored as MG-tied IN or MG-tied PN, and IN and PN cell bodies that are removed by more than one MG cell body length are scored MG-free IN and MG-free PN (*Figure 5B*). Overall, as expected, MGs are more prevalent at older ages (*Figure 5C*). We then compared the cytoplasmic POLK levels between the MG-tied IN and MG-free INs as well as MG-tied and MG-free PNs. Interestingly, early-old MG-tied INs have significantly more cytoplasmic POLK granules intensity and counts compared to MG-free INs and MG-free PNs (*Figure 5D*, *Figure 5—figure supplement 1*). While the mechanism is unknown, we speculate loss of MG-tied INs in late-old age most likely reflects a survivorship bias, perhaps due to the MG-mediated removal of the INs from the early old stage.

**Table 8.** Cell class nuclear Polk counts per age and brain area.

**Post hoc comparisons – age_bracket × cell type × area**

| | | Mean Diff | SE | t | Cohen's d | p_tukey | |
|---|---|---|---|---|---|---|---|
| Young IN M1 | Young PN M1 | 6.422 | 1.891 | 3.396 | 0.528 | 0.034 | * |
| Young IN M1 | Young NN M1 | 29.677 | 1.872 | 15.854 | 2.44 | <0.001 | *** |
| Young PN M1 | Young NN M1 | 23.255 | 1.079 | 21.549 | 1.912 | <0.001 | *** |
| Young IN S1 | Young PN S1 | 8.764 | 1.715 | 5.111 | 0.721 | <0.001 | *** |
| Young IN S1 | Young NN S1 | 29.129 | 1.704 | 17.094 | 2.395 | <0.001 | *** |
| Young PN S1 | Young NN S1 | 20.366 | 0.956 | 21.297 | 1.675 | <0.001 | *** |
| Early old IN M1 | Early old PN M1 | 18.541 | 1.997 | 9.282 | 1.525 | <0.001 | *** |
| Early old IN M1 | Early old NN M1 | 28.156 | 1.945 | 14.474 | 2.315 | <0.001 | *** |
| Early old PN M1 | Early old NN M1 | 9.615 | 1.159 | 8.296 | 0.791 | <0.001 | *** |
| Early old IN S1 | Early old PN S1 | 15.842 | 1.935 | 8.185 | 1.303 | <0.001 | *** |
| Early old IN S1 | Early old NN S1 | 32.222 | 1.878 | 17.162 | 2.649 | <0.001 | *** |
| Early old PN S1 | Early old NN S1 | 16.38 | 0.988 | 16.571 | 1.347 | <0.001 | *** |

Note. p-value adjusted for comparing a family of 12, only relevant comparisons shown. *p < 0.05, **p < 0.01, ***p < 0.001.

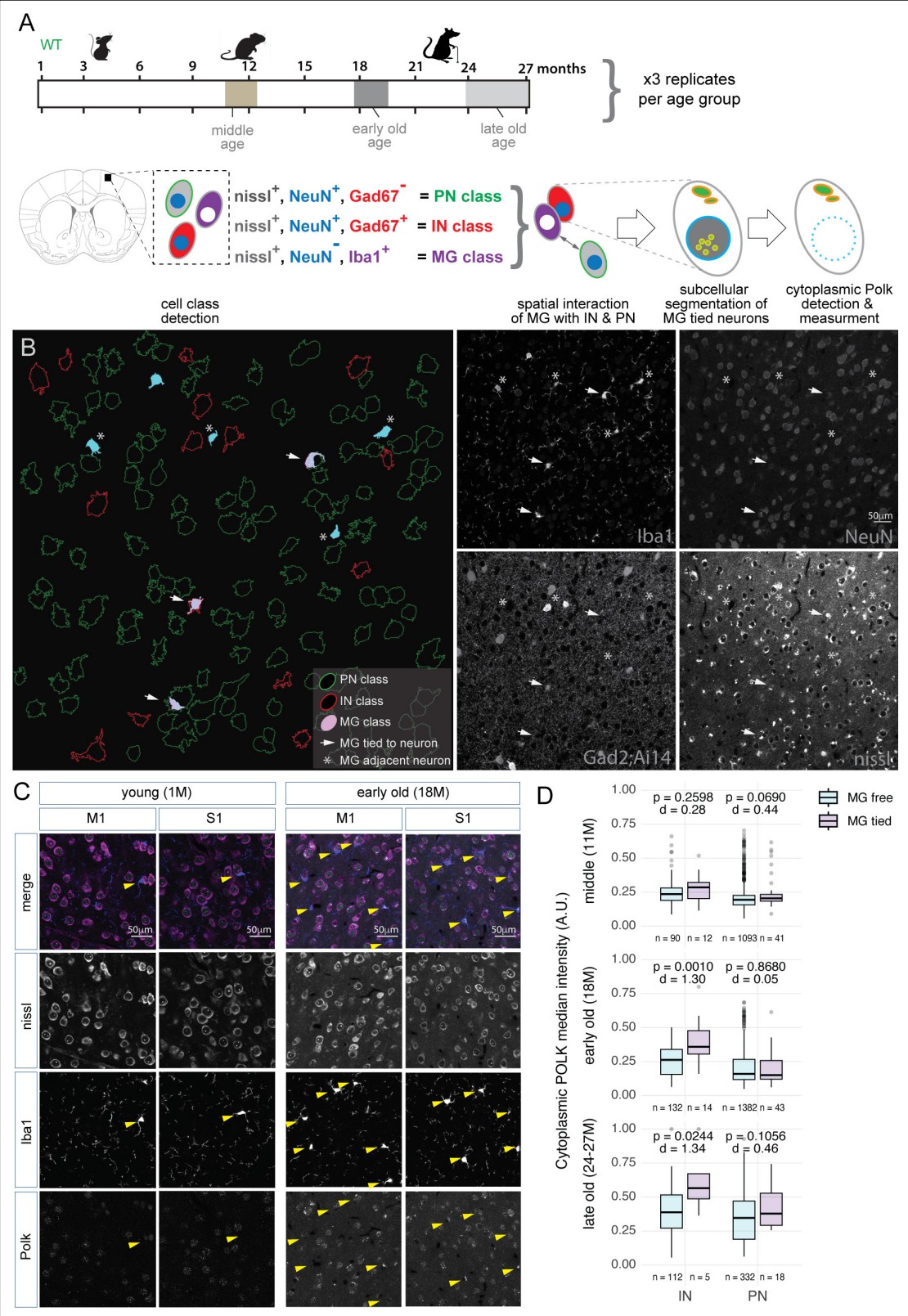

**Figure 5.** Microglia associated with INs and PNs show significantly higher levels of cytoplasmic POLK expression. (**A**) Schematic of the experimental design showing age groups, cell class gating logic, scoring spatial interaction, subcellular segmentation, detection, and measurement of POLK in the cytoplasmic compartment. (**B**) A representative single 20x imaging field from Gad2; Ai14 brain shows outlines of detected cell bodies of IN (red), PN (green). Iba1+ microglia (MG) that are attached or wrapped to neurons (MG-tied, filled light pink and arrows) and microglia that are within one

*Figure 5 continued on next page*

*Figure 5 continued*

body length of a neuron (blue fill with an asterisk) as well as unassociated microglia (blue fill). Channel separation shown for the four-channel super-resolution confocal image, fluorescent-nissl, NeuN, and Gad2; Ai14 = IN, fluorescent-nissl, and NeuN = PN, and fluorescent-nissl, and Iba1 = MG. (**C**) Representative images from M1 and S1 cortical areas show an overall increase in Iba1-positive cells in early old age compared to young. Yellow arrowheads point to MG-tied neurons. (**D**) Boxplot showing the difference in cytoplasmic POLK granule median intensity between MG-free and MG-tied INs and PNs across middle-, early-, and late-old time age groups. *T*-test p-values and effect sizes are shown for each comparison. Images were quantitated on fluorescent-nissl, NeuN, and Gad67 = IN, fluorescent-nissl, and NeuN = PN, and fluorescent-nissl, and Iba1 = MG.

The online version of this article includes the following source data and figure supplement(s) for figure 5:

**Source data 1.** CSV file and R script for boxplots comparing the nuclear POLK median intensity and cytoplasmic POLK granule count.

**Figure supplement 1.** Boxplots comparing the nuclear POLK median intensity and cytoplasmic POLK granule count between MG-free and MG-tied INs and PNs across middle, early-old, and late-old time age groups.

Such differences were not noted in nuclear POLK counts between MG-tied and MG-free INs and PNs (*Figure 5—figure supplement 1*). Combined with the observations of elevated DNA damage and repair in INs (*Figure 2*), and higher cytoplasmic POLK levels in MG-tied INs in early old age, suggesting potential increased vulnerability of IN class during aging.

## Subcellular localization of POLK is regulated by neuronal activity

Neuronal activity is essential for proper neuronal maturation and circuit plasticity (*Yap and Greenberg, 2018*). High metabolic demands during increased neuronal activity can generate oxidative damage to actively transcribed regions of the genome (*Lu et al., 2004*). Neuronal activity-induced transcription has been linked with the generation of repeated DSBs at gene regulatory elements, potentially contributing to genome instability (*Welch and Tsai, 2022*; *Suberbielle et al., 2013*; *Madabhushi et al., 2015*; *Delint-Ramirez et al., 2022*; *Crowe et al., 2006*). Although the coupling of transcription to DNA breaks occurs in all cell types, this process is especially challenging to long-lived neurons that are postmitotic and cannot use replication-dependent DNA repair pathways. With limited regenerative capacity, neurons are unable to replace the damaged cells (*Iyama and Wilson, 2013*). Recently, a neuronal-specific complex NPAS4–NuA4 was identified to be bound to recurrently damaged regulatory elements and recruits additional DNA repair machinery to stimulate the repair, thus coupling neuronal activity to DNA repair (*Pollina et al., 2023*).

Since we observed a decrease in nuclear POLK and concomitant increase in cytoplasmic POLK with age, as well as nuclear POLK to be associated with BER and NHEJ DNA repair pathways in the mice brain, we tested if POLK subcellular localization can be regulated by neuronal activity and if this regulation is influenced by age. For this, we tested two age groups, young (1–2 month) and early-old (18 month) brains. Following acute slice electrophysiology protocol, mouse brains were extracted, and fresh tissue was sectioned coronally and immersed in oxygenated artificial cerebrospinal fluid (ACSF) and maintained at physiological temperature. Coronal sections from the young or early-old brains were split into two hemispheres. One hemisphere was maintained in ACSF only forming the control group, and the other hemisphere was exposed to a bath application of kainic acid (KA), a glutamate receptor agonist that synchronously depolarizes neurons, dissolved in ACSF, which was the treatment group. Each hemisphere either in control ACSF or ACSF + KA were exposed for a duration of 80 and 160 min. After 80 or 160 min the hemispheres were removed and quickly fixed in 4% paraformaldehyde (PFA) to terminate the experiment. The fixed slices were immunolabeled for NeuN, fluorescent nissl, c-FOS, and POLK to measure changes in c-FOS and POLK protein levels in neurons by IF (*Figure 6A*). For each age group and exposure (80 and 160 min), we ran a non-parametric two-sided Mann–Whitney *U* test on the dependent variables and computed Cohen's *d* (KA–CTRL) for effect size.

c-FOS protein is known to be detected by 90 min sustaining for 2–5 hr upon neuronal activity (*Hudson, 2018*; *Kovács, 1998*), and we observed a robust increase in c-FOS intensity in the nucleus at 160 min in both young and early-old age groups (*Figure 6B*). Interestingly, at 160 min we observed a significant increase in nuclear POLK speckle counts (*Figure 6C*) and a concomitant decrease in cytoplasmic POLK granules (*Figure 6D*) in young mice brains. However, such changes in POLK levels were not observed in early-old brains (*Figure 6C, D*), indicating inducing neuronal activity can elicit a nuclear shift of POLK in young brains which fails in early-old brains. Since we did not distinguish between IN, PN, and NN classes, we cannot exclude the possibility of even higher differences in activity-induced POLK subcellular localization to the nucleus among different cell classes.

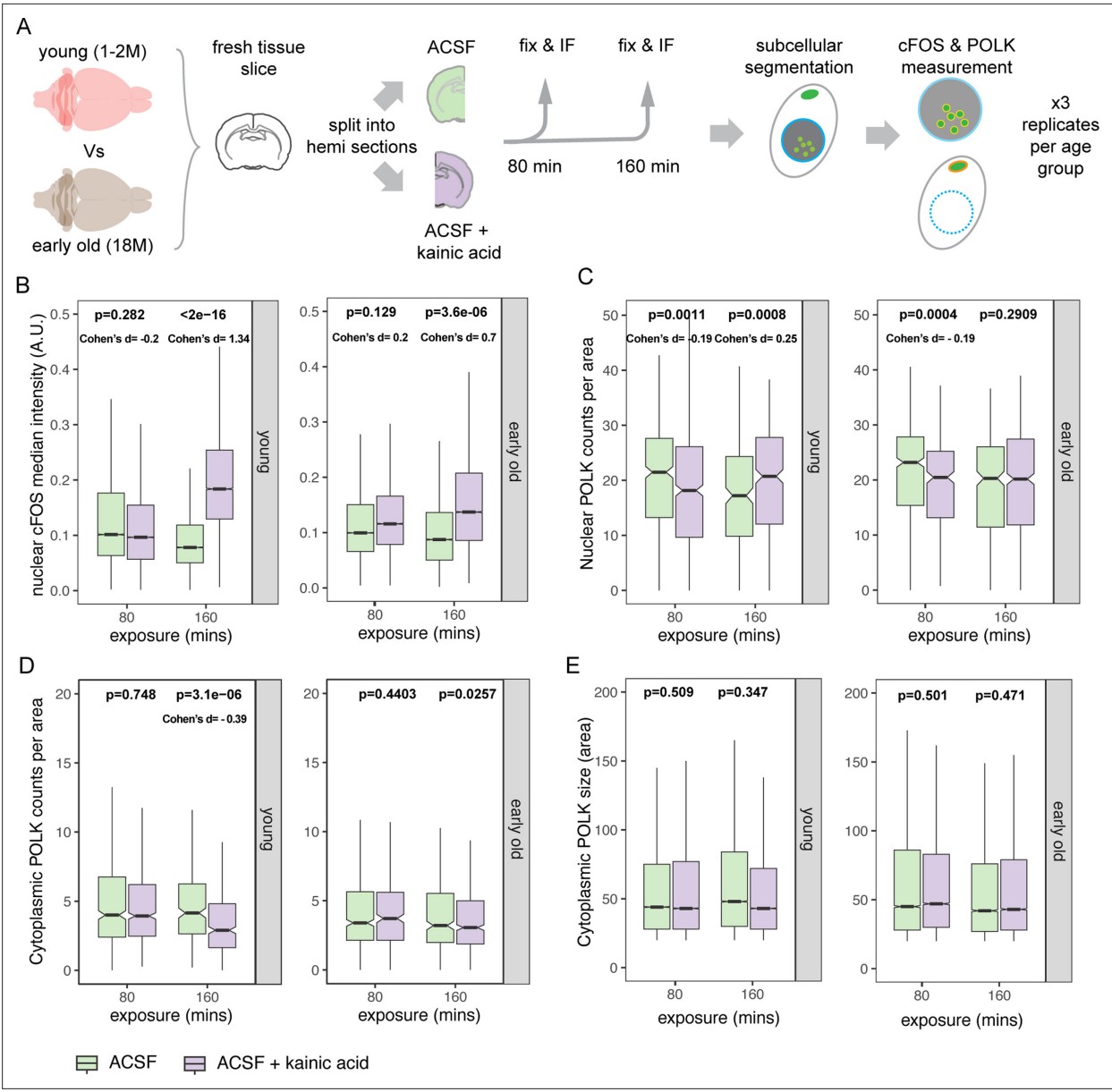

**Figure 6.** Kainic acid-induced neuronal activity causes POLK subcellular localization in young ex vivo brain. (**A**) Schematic of the experimental design showing age groups, ex vivo treatment of kainic acid, and control groups, followed by immunostaining, imaging, and the measurement of nuclear and cytoplasmic POLK. (**B**) Boxplot of c-FOS levels between artificial cerebrospinal fluid (ACSF) control and kainic acid-treated groups at 80 and 160 min post-exposure intervals between young and old brains. Due to large number of observations artificially influencing significance, data was randomly subsampled 200 times, each time with 100 cells per group for Wilcoxon rank-sum (Mann–Whitney $U$) two-sided test. Distribution of subsampled p-values and Cohen's $d$ are shown in **Figure 6—figure supplement 1A**. Median p-value and Cohen's $d$ (gray columns in **Figure 6—figure supplement 1A**) are reported on the plot. (**C**) Boxplot of nuclear POLK speckle counts per unit between ACSF control and kainic acid-treated groups at 80 and 160 min post-exposure intervals between young and old brains. Wilcoxon rank-sum (Mann–Whitney $U$) two-sided test showed a significant increase in nuclear POLK at 160 min in the young brains. (**D**) Boxplot of cytoplasmic POLK granule counts per unit area between ACSF control and kainic acid-treated groups at 80 and 160 min post-exposure intervals between young and old brains. Wilcoxon rank-sum (Mann–Whitney $U$) two-sided test shows there is a significant decrease with small effect size in cytoplasmic granule counts upon kainic acid treatment after 160 min in young brains. (**E**) Boxplot of cytoplasmic POLK granule size measured by area contained, between ACSF control and kainic acid-treated groups at 80 and 160 min post-exposure intervals between young and old brains. Due to large number of observations artificially influencing significance, data was randomly subsampled 200 times, each time with 100 cells per group for Wilcoxon rank-sum (Mann–Whitney $U$) two-sided test. Distribution of subsampled p-values and Cohen's $d$ are shown in **Figure 6—figure supplement 1B**. Median p-value and Cohen's $d$ (gray columns in **Figure 6—figure supplement 1B**) are reported on the plot.

The online version of this article includes the following source data and figure supplement(s) for figure 6:

**Source data 1.** CSV files, R and Python script for subsampling and distribution of p-values and Cohen's d.

**Figure supplement 1.** Subsampling distribution of p-values and Cohen's d.

Acute brain slice experiments are a powerful assay system for studying synaptic plasticity, circuit organization, and molecular and cellular mechanisms in ex-vivo rodent brains (*Petreanu et al., 2009*, PMID: 19151697; *Lo et al., 1994*, PMID: 7993619; *Svoboda and Yasuda, 2006*, PMID: 16772166). However, the process of acute brain slices is prolonged and quite different compared to the rapid in-vivo fixation of proteins in an intact brain by vascular PFA perfusion that freezes the cell-molecular milieu. Acute slices impose stressors like transient hypoxia, mechanical trauma of cutting, and initial metabolic shock that can induce c-FOS, CCAAT enhancer binding protein (C/EBP), and brain-derived neurotrophic factor (*Taubenfeld et al., 2002*) and may cause redox damage (*Malkov et al., 2014*; *D'Agostino et al., 2007*; *Sasaki et al., 2018*), especially in old brains (*Ting et al., 2014*). Hence directly comparing baseline nuclear POLK in intact brains to ex-vivo slices exposed to prolonged oxygenated ACSF may be inappropriate, as both the young and old slices are in a shared artificial state. This may explain the neuronal POLK baseline levels for young and old being similar in the ex-vivo assay (*Figure 6C*). Despite these limitations, a timed exposure of acute slices from the same animal to either KA or ACSF provided a uniquely controlled observational opportunity to ask a bigger question, if a Y family TLS polymerase can at all respond to neuronal activity and within what time scale. Our results provide first direct evidence of POLK shuttling upon neuronal activity within hours, likely to retain it in the nucleus for activity-induced DNA repair process, a capacity that was absent in older brains.

We, however, do not have an explanation for the surprising temporary initial dip in nuclear POLK puncta seen in both young and old ex-vivo slices only upon KA treatment at 80 min with very similar magnitudes, Cohen's $d = -0.19$ (*Figure 6C*). Since this activity-induced observation is age independent, it may indeed be a hint to an entirely novel biological phenomenon of POLK (and perhaps TLS polymerases in general) that occurs at a rapid time scale which would not otherwise have been captured in a perfused fixed tissue end-point assay. This will need a detailed investigation using continuous fluorescence imaging of live tissues that is beyond the scope of this current study.

## POLK nucleo-cytoplasmic status is a learnable feature of neuronal age

There have been multiple approaches toward estimating biological age using 'aging clocks' by sequencing, from tissue and body fluids using epigenetic marks (*Grodstein et al., 2020*) and multi-omics approaches (*Nie et al., 2022*). Further genomic instability is increasingly associated with neuronal activity and aging, with Progeroid syndromes often linked to compromised genomic integrity (*Gurkar and Niedernhofer, 2015*). However, there is a lack of in situ methods to determine age from tissues. Hence, we hypothesized that it may be possible to leverage the protein expression of POLK to determine the biological age of brain tissue sections. Specifically, we tested if the naturally occurring shift in the subcellular localization of the endogenous POLK can be a learnable feature that can predict organismal age from brain sections. For this, we used two related non-parametric ensemble-based classifiers, Random Forest and Gradient Boosting Machine, where 80% of the data was used for training and validation, and then tested on a 20% hold-out group. Both learning methods are based on recursive partitioning of the feature space through decision trees. Unlike parametric statistical tests, they do not assume normality, homoscedasticity, or linear relationships between predictors and outcome variables. Instead, these models assume that (1) the input features are relevant and measured without systematic bias, (2) the samples are independent and identically distributed, and (3) the training data sufficiently represent the underlying population to avoid overfitting. Random Forest reduces variance through bagging (bootstrap aggregation) and random feature selection, producing an average prediction across many decorrelated trees. Gradient Boosting, in contrast, sequentially builds trees that correct errors made by prior iterations, minimizing a differentiable loss function and often achieving higher sensitivity at the cost of greater susceptibility to overfitting. Feature importance was quantified using total increase in node purity (Random Forest) and relative influence (Gradient Boosting), both reflecting the average improvement in model fit attributable to each predictor variable.

We first used the smaller dataset from *Figure 1* and parameters like nuclear and cytoplasmic POLK counts, nuclear area, cytoplasmic area, and replicates were tested to examine if organismal age is discernible. Further, using a larger dataset and parameters from *Figure 5*, where we had observed the association of MG with increased cytoplasmic POLK in IN-class, we tested if finer distinctions between middle, early-old, and late-old age groups are possible. In the smaller dataset (*Figure 7* Panels A1, A2,

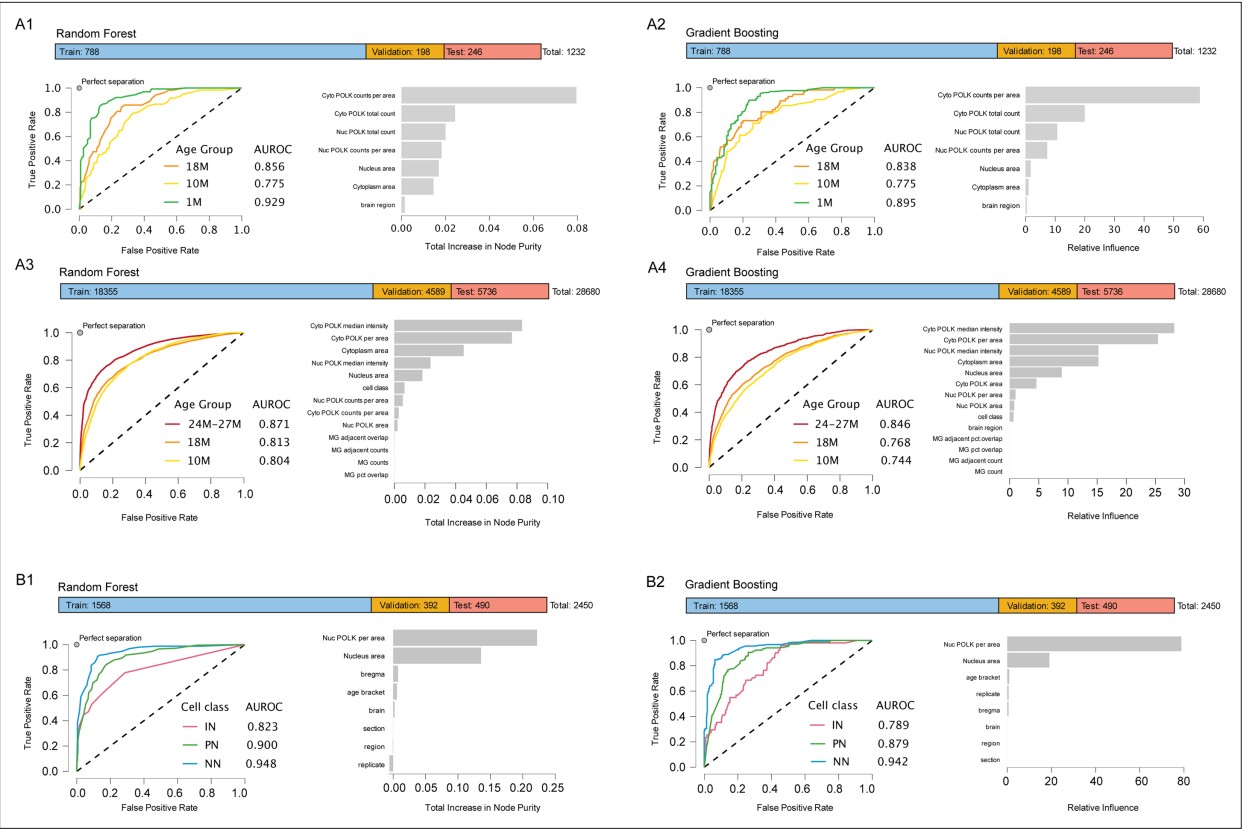

**Figure 7.** Subcellular expression of POLK is a learnable feature predictive of organismal age and IN, PN, and NN cell class from mouse brain tissue. Random Forest (**A1**) and Gradient Boosting (**A2**) classifier can distinguish between 1-, 10-, and 18-month-old age groups upon training on the subcellular nuclear and cytoplasmic distribution of POLK counts using data from *Figure 1F, G*. Top bar shows the number of cells used for training, validation, and test samples of the total. Area Under Receiver Operator Curve (AUROC) values show the performance of the classifier for each age group when tested on the holdout groups. The bar plot shows the relative contribution of the image parameters and image metadata indicating cytoplasmic POLK counts as the major driver. Random Forest (**A3**) and Gradient Boosting (**A4**) classifier can distinguish between middle to late-old ages 10-, 18-, and 24- to 27-month age groups upon training on the subcellular nuclear and cytoplasmic distribution of POLK counts, POLK intensities in nuclear speckles and cytoplasmic granules, MG association and other metadata from *Figure 5*. Top bar shows the data split for training, validation, and test groups. Major driver of performance is cytoplasmic POLK intensity and counts. Random Forest (**B1**) and Gradient Boosting (**B2**) classifier can distinguish between broad IN, PN, and NN cell class with high AUROC values. Nuclear POLK per unit area and nuclear area are the major feature drivers of performance.

The online version of this article includes the following source data for figure 7:

**Source data 1.** CSV files and JASP analysis files for subcellular expression of POLK is a learnable feature predictive of organismal age and IN, PN, and NN cell class.

*N* = 1232; Train = 788, Validation = 198, Test = 246), both models achieved strong classification performance. The Random Forest model yielded Area Under Receiver Operator Curve (AUROC) values of 0.929, 0.775, and 0.856 for 1-, 10-, and 18-month groups, respectively, while the Gradient Boosting model achieved comparable AUROC values of 0.895, 0.775, and 0.838. Across both algorithms, the most influential predictors were cytoplasmic POLK counts per area, cytoplasmic POLK total count, nuclear POLK total count, and nuclear POLK counts per area, followed by morphometric features such as nucleus area and cytoplasm area. (*Figure 7A3, A4*). In the larger dataset (*Figure 7* Panels A3–A4, N=28,680; Train = 18,355, Validation = 4,589, Test = 5,736), model performance remained robust. The Random Forest classifier produced AUROC values of 0.804 (10M), 0.813 (18M), and 0.871 (24–27M), while the Gradient Boosting classifier yielded 0.744, 0.768, and 0.846, respectively. In both models, cytoplasmic POLK median intensity, cytoplasmic POLK per area, nuclear POLK median intensity, and cytoplasmic area were the most important features.

Following a similar strategy, using data from young and old groups shown in *Figure 4*, we observed that in both age groups nuclear POLK speckle counts per unit area alone are the major driver in

distinguishing IN, PN, and NN classes with ROC >0.8 (*Figure 7B1, B2*), suggesting cell classes IN, PN, and NN have an inherent biological requirement in how they utilize TLS polymerase POLK. Together, these analyses indicate that quantitative measures of POLK localization, particularly cytoplasmic intensity and spatial distribution, can potentially serve as reliable predictors of age-dependent cellular phenotypes. The consistent ranking of variable importance across both modeling approaches supports the robustness of these features in distinguishing neuronal aging stages. Here, we recognize that as a terminal assay, POLK nucleo-cytoplasmic status is impractical for longitudinal studies. Additional in-depth molecular mechanistic studies of POLK subcellular shuttling in healthy aging, in neurodegenerative models, and upon gerotherapeutic treatments are needed. However, we believe this observed age association may fill a useful niche as an endogenous in-situ readout of brain tissue age, given the scarcity of such cell-biological markers compared to cell culture systems.

## Discussion

TLS polymerases are well-studied as DNA lesion bypass proteins in dividing cells, and recent extensive research identifies their novel roles in a myriad of other cellular processes (*Anand et al., 2023*; *Paniagua and Jacobs, 2023*). Despite evidence of mRNA expression of the Y-family TLS polymerase *Polk* in IN and PN subtypes (*Tasic et al., 2018*; *Paul et al., 2017*), the role of POLK in postmitotic cells has not yet been explored in the central nervous system (CNS). There is only one study of POLK in the peripheral nervous system, where, upon cisplatin treatment, *Polk* mRNA was upregulated in DRG neurons (*Zhuo et al., 2018*), and POLK was found essential for efficiently and accurately repairing cisplatin crosslinks (*Jha and Ling, 2018*). In a cell-free system, POLK faithfully bypasses 8oxo-dG (*Maddukuri et al., 2014*), which is a major DNA lesion in the neurons. Interestingly, Kaplan–Meier survival curves of *Polk*$^{-/-}$ mice showed decreased survival starting at 1 year compared to wild-type (WT) and heterozygous littermates, and increased levels of spontaneous mutations (*Stancel et al., 2009*). Separately, inactivated *Polk* knock-in mice showed a higher frequency of mutations, micronucleated cells, and DNA damage marker gH2AX foci compared to WT mice (*Takeiri et al., 2014*). In these studies, the effect of *Polk*$^{-/-}$ was not explored in the brain. However, *Apoe−/−;Polk−/−* mice showed enhanced DNA mutagenesis in several organs, including the brain, probably due to cholesterol-induced adducts (*Singer et al., 2013*).

Given that DNA damage markers 8oxo-dG and gH2AX are highly correlated with aging, and Polk is associated with multiple DNA repair mechanisms, as well as POLK can function in non-S phase cells (*Ogi and Lehmann, 2006*; *Sertic et al., 2018*), however, it remained unknown if normative age-associated DNA damage will also recruit POLK and how it may function in postmitotic neurons. This is the first systematic and longitudinal study of Y-family TLS polymerase focusing mostly on the POLK in postmitotic neurons. Here, we found that Y-family TLS polymerases, including POLK, are highly expressed in several cortical brain areas with characteristic nuclear speckle-like distribution, reminiscent of nuclear foci in dividing cells (*Bi et al., 2005*; *Bergoglio et al., 2002*, 1). DNA repair proteins are often found to form dynamic membrane-less biomolecular condensates or 'foci' at damaged sites in response to DSBs; the formation of these foci within the proper time is essential for maintenance of genomic integrity. An emerging hypothesis is that the repair condensates are formed via liquid–liquid phase transition (*Hyman et al., 2014*; *Miné-Hattab et al., 2022*; *Banani et al., 2017*). From our human cell line iPoKD-MS datasets (*Paul et al., 2023*), there are also hints of POLK's association with proteins of other nuclear biomolecular condensates like paraspeckles, Cajal bodies, PML nuclear bodies, polycomb (PcG) bodies, and the nucleolus. Hence, we speculate that POLK speckles in the neuronal nucleus are also dynamically localized in various biomolecular condensates depending on context that might facilitate DNA repair and other unknown functions.

Interestingly, we noted that brain nuclear lysate shows two distinct POLK bands (~120 and ~99 kDa), with an enrichment of the lower mobility band ~120 kDa and absence in the cytoplasmic fractionation. We observed a similar pattern of bands that were present differentially in the chromatin fraction versus the cytoplasmic fraction in human iPSC-derived neurons and multiple other human cancer cell lines that were reduced upon knocking down POLK (unpublished data). Although the 99 kDa is the reported POLK band in literature, we do not exactly understand the nature of the higher molecular weight POLK proteins present in the cellular fractionations. We have a few hypotheses based on previous literature on other Y-family polymerase-like POLH proteins, as well as our iPoKD-MS data of the POLK interactome in POLK active genomic sites. POLH has been identified to be

post-translationally modified (PTM), such as human POLH undergoing O-GlcNAcylation by O-GlcNAc transferase upon DNA damage (*Ma et al., 2017*), multisite SUMOylation of POLH being essential for the removal of POLH from the DNA damage sites (*Guérillon et al., 2020*), and Rad18-dependent SUMOylation of POLH being required to target it to the replication fork and prevent under-replicated DNA (*Despras et al., 2016*). From the existing knowledge, it will be important to identify the PTMs of POLK in the neurons and how these modifications assist in POLK subcellular localization and activity. We predict that either the higher molecular weight (potentially modified POLK or another isoform of POLK) is absent in the cytoplasmic fraction, or the fractionation procedure is unable to extract the higher molecular weight POLK from particular cytoplasmic organelles, like the endo-lysosomes.

Even in the absence of any exogenous chemical or behavioral stressors, in wild-type mice, POLK in neurons localizes to the cytoplasm at least as early as 10–11 months (middle age) and continues to do so with further normative aging. In fact, this nuclear to cytoplasm POLK shuttling could train two ensemble learning models to predict the age of the mouse brain, where cytoplasmic and nuclear POLK counts were the primary featured drivers. Both learning models performed comparatively well, with possible applications as in situ IF-based indicators of age-dependent cellular phenotypes. This can be complementary to existing sequencing-based epigenetic clocks (*de Lima Camillo et al., 2022*; *Bell et al., 2019*; *Prosz et al., 2024*; *Griffin et al., 2024*; *Varshavsky et al., 2023*; *Kerepesi et al., 2021*), allowing simultaneous visual read-out of brain tissue age while scoring for various markers for neurodegeneration, cell stress, and age-associated neuropathologies. We believe other TLS proteins may behave similarly, and a combinatorial approach can even increase the age-predictive power. We were able to make even finer distinctions among middle-, early-, and late-old stages using the cytoplasmic POLK intensity signal. We predict that multiple stressors during the process of neuronal aging are inducing a cytoplasmic shift of POLK, which needs to be further explored to gain mechanistic understanding of neuronal POLK activity.

Upregulation, stabilization, and subcellular shuttling of POLK have been reported earlier in human cancer cells. Upon oncogene inhibition in melanoma, lung, and breast cancer cells, POLK expression was increased and localized from cytoplasm to nucleus, leading to resistance to the oncogene inhibiting chemotherapy drugs. Treatment with leptomycin B, an exportin-1 inhibitor, increased the accumulation of nuclear POLK, suggesting oncogenic signaling partially regulates nuclear POLK through the export machinery. A similar nuclear localization of POLK also occurred upon inhibition of the kinase mTOR, by induction of ER stress, or by glucose deprivation, as well as rapid export of POLK to the cytoplasm after stress, suggesting dynamic regulation of POLK upon stress. Interestingly, the upregulation and nuclear localization were found not excessively mutagenic (*Temprine et al., 2020*). In addition, upon carcinogen-induced nucleolar stress, cellular POLK protein stability increased with enrichment of POLK and its activity in the nucleolus, and was identified to play a critical role in recovery from nucleolar stress (*Paul et al., 2023*). While POLK overexpression in cancer cells is known to drive mutagenesis and chemoresistance, however, its increased stability to recover from nucleolar stress and other additional stress (*Temprine et al., 2020*; *Paul et al., 2023*), suggests POLK can multitask based on cellular context, which can be dependent on its catalytic or non-catalytic functions.

Our findings showing progressive loss of nuclear POLK and accumulation in the cytoplasm suggest nuclear POLK plays an important role in protecting neurons from DNA damage at a young age. Decline of nuclear POLK is likely correlated with lack of repair leading to DNA damage accumulation during the normative aging process. Our data further confirm nuclear POLK is associated with neuronal DNA damage as observed by colocalization with DSB (gH2AX) and ROS-mediated DNA damage sites (8oxo-dG), as well as POLK's role in the BER (colocalization with APE1 and LIG3) and NHEJ repair pathways (53BP1 and PRKDC) in the neurons.

An enduring question revolves around the differential vulnerability of neuronal subtypes to DNA damage (*Welch and Tsai, 2022*). Remarkably, we observed POLK's nuclear expression to be cell class-specific, with the highest levels in INs, followed by PN and NN. INs and PNs engage in highly demanding cellular functions, both transcriptionally and energetically. While other important and more abundant cell types in the nervous system, such as astrocytes, oligodendrocytes, and microglia, collectively grouped in this study as NNs, are also vulnerable to DNA damage, their characteristics differ significantly from neurons. For instance, NNs can be dispensed as they are largely renewable, possess relatively lower energy demands due to slower physiological kinetics, and can re-enter the cell cycle when necessary (*Welch and Tsai, 2022*). These attributes are thought to collectively lessen the

dependence on DNA damage repair in glial cells compared to neurons. Hence, the observed lowest nuclear expression of POLK in NNs, at face value, can be reconciled with their known biology. On the other hand, a significant fraction of the IN types, like PV basket cells, Chandelier cells, and Martinotti cells across cortical areas, have fast-firing kinetics and hence are energetically more demanding (*Markram et al., 2004*; *Tremblay et al., 2016*); arguably, accumulating more damage and DNA adducts may require the higher nuclear POLK presence. Here too, the disparity of nuclear POLK was reproducible enough across young and older ages that measuring the nuclear POLK speckle counts per unit area can predict cell class identity using ensemble learning with consistently high AUROCs.

The association between neurons harboring DNA damage, age-associated neuroinflammation, and cell-type vulnerability is an important piece of the puzzle to understand age-associated neurodegeneration under normal physiological conditions (*Welch and Tsai, 2022*). We observed progressive accumulation of cytoplasmic POLK in SGs and endo/lysosomal compartments with age, possibly due to impaired proteostasis. The colocalization of cytoplasmic gH2AX and POLK granules indicates association with CCF and will be important to understand POLK's role in driving inflammaging.

Interestingly, only microglia-associated INs had significantly higher cytoplasmic POLK, suggesting they are more vulnerable to cumulative age-associated changes. One possibility is that the decline in nuclear POLK in INs causes a reduction in DNA repair that may further initiate cGAS-STING immune signaling, thereby recruiting microglia (*Gulen et al., 2023*; *Talbot et al., 2024*), or higher levels of cytoplasmic POLK, coupled with cytoplasmic gH2AX, may itself lead to immune activation. Further in-depth molecular mechanistic studies are needed to dissect POLK's role in neuronal cell class-specific genome maintenance and how it intersects with microgliosis during aging.

Another factor contributing to the heightened vulnerability of neurons to genomic damage is their involvement in functional processes like neuronal activity. Such as induced activity of primary neurons with bicuculline or exposure of mice to fear learning was shown to generate DSBs at the promoters of immediate-early genes (*Madabhushi et al., 2015*; *Stott et al., 2021*), and even mice experiencing a new environment can induce DSBs in neurons (*Suberbielle et al., 2013*). These activity-induced DSBs are thought to aid in the expression of immediate early genes by swiftly resolving topological constraints at their transcription start sites. Here, we showed inducing neuronal activity in brain slices increases nuclear POLK while diminishing cytoplasmic accumulation. While it is an artificial ex-vivo scenario, this approach shows that POLK, at least in principle, is responsive to induced neuronal activity, and we speculate such an increase in nuclear POLK is likely to be involved in activity-induced DNA repair. However, we noticed that this modulatory effect is lost upon aging, indicating some yet unknown mechanism at play that remains to be elucidated. Thus, in our study, we provide first-time evidence of Y-family TLS polymerase, POLK's differential expression in CNS cell classes, and its age-associated and activity-induced subcellular changes, with implications in microgliosis under non-pathogenic aging conditions.

## Methods

**Key resources table**

| Reagent type (species) or resource | Designation | Source or reference | Identifiers | Additional information |
|---|---|---|---|---|
| Antibody | Mouse monoclonal anti-PolK (or) DinB Antibody (A-9) | Santa Cruz | RRID:AB_2167029 | IF (1:200 and 1:400) WB (1:500) |
| Antibody | Mouse monoclonal anti-PolK (or) DinB Antibody (A-9) HRP | Santa Cruz | sc-166667 HRP | WB (1:500) |
| Antibody | Rabbit polyclonal anti-PolK | ABclonal | RRID:AB_2758963 | IF (1:8000) WB (1:2000) |
| Antibody | Mouse monoclonal anti-REV1 | Santa Cruz | RRID:AB_2885169 | IF (1:400) |
| Antibody | Mouse monoclonal anti-Poli | Santa Cruz | RRID:AB_2167019 | IF (1:400) |
| Antibody | Guinea Pig polyclonal anti-NeuN/Fox3 | Synaptic Systems | RRID:AB_2619988 | IF (1:500) |
| Antibody | Mouse monoclonal anti-Gad67 | Millipore | RRID:AB_2278725 | IF (1:1000) |
| Antibody | Rabbit polyclonal anti-Gamma h2ax | Novus Biologicals | RRID:AB_350295 | IF (1:2000) |

*Continued on next page*

*Continued*

| Reagent type (species) or resource | Designation | Source or reference | Identifiers | Additional information |
|---|---|---|---|---|
| Antibody | Rabbit polyclonal anti-53BP1 | Novus Biologicals | RRID:AB_350221 | IF (1:2000) |
| Antibody | Rabbit polyclonal anti-8-OHdG | GeneTex | RRID:AB_2893390 | IF (1:100) |
| Antibody | Rabbit polyclonal anti-DNA-PKcs | ABclonal | RRID:AB_2771830 | IF (1:100) |
| Antibody | Rabbit polyclonal anti-XRCC4 | ABclonal | RRID:AB_2861842 | IF (1:100) |
| Antibody | Rabbit polyclonal anti-DNA Ligase III | ABclonal | RRID:AB_2760369 | IF (1:100) |
| Antibody | Rabbit polyclonal anti-Ku70 | ABclonal | RRID:AB_2861526 | IF (1:100) |
| Antibody | Rabbit polyclonal anti-XRCC1 | ABclonal | RRID:AB_2757194 | IF (1:100) |
| Antibody | Rabbit polyclonal anti-Ape1 | ABclonal | RRID:AB_2861531 | IF (1:100) |
| Antibody | Rabbit polyclonal anti-Iba1 | Abcam | RRID:AB_2636859 | IF (1:1000) |
| Antibody | Mouse monoclonal anti-G3BP1 | DSHB | RRID:AB_2722179 | IF (1:18) |
| Antibody | Rat monoclonal anti-1D4B (LAMP1) | DSHB | RRID:AB_2134500 | IF (1:200) |
| Antibody | Rabbit polyclonal anti-cFOS | Synaptic Systems | RRID:AB_2891278 | IF (1:1000) |
| Antibody | Mouse monoclonal anti-Alpha Tubulin | Proteintech | RRID:AB_2687491 | WB (1:10,000) |
| Antibody | Mouse monoclonal anti-Beta Actin | Proteintech | RRID:AB_2819183 | WB (1:10,000) |
| Antibody | Mouse monoclonal anti-GAPDH (G9) | Santa Cruz | RRID:AB_10847862 | WB (1:2000) |
| Antibody | Mouse Histone-H3 (1G1) | Santa Cruz | RRID:AB_2848194 | WB (1:1000) |
| Antibody | Chicken polyclonal anti-MAP2 | Abcam | RRID:AB_2138153 | IF (1:10,000) |
| Antibody | Rabbit polyclonal anti-Cathepsin B | ProteinTech | RRID:AB_2086929 | IF (1:500) |
| Antibody | Rabbit polyclonal anti-Cathepsin D | Proteintech | RRID:AB_10733646 | IF (1:500) |
| Antibody | Rabbit polyclonal anti-Glococerebrosidase (GBA) | Sigma-Aldrich | RRID:AB_1078958 | IF (1:500) |
| Antibody | Rabbit polyclonal anti-EEA1 | ProteinTech | RRID:AB_2881117 | IF (1:500) |
| Antibody | Goat polyclonal Anti-Mouse IgG(H+L) HRP-conjugated | ProteinTech | RRID:AB_2722565 | WB (1:10,000) |
| Antibody | Goat anti-Guinea Pig Dylight 405 | Jackson ImmunoResearch | RRID:AB_2337432 | IF (1:1000) |
| Antibody | Goat anti-Mouse IgG1 Alexa Fluor 488 | Invitrogen | RRID:AB_2535764 | IF (1:1000 and 1:2000) |
| Antibody | Donkey anti-Mouse IgG Alexa Fluor 488 | Invitrogen | RRID:AB_2762838 | IF (1:1000) |
| Antibody | Goat anti-Mouse IgG2a Alexa Fluor 594 | Invitrogen | RRID:AB_2535774 | IF (1:1000) |
| Antibody | Goat anti-Rat Alexa Fluor 594 | Invitrogen | RRID:AB_2896333 | IF (1:1000) |
| Antibody | Goat anti-Rabbit Alexa Fluor 647 | Invitrogen | RRID:AB_2633282 | IF (1:1000) |
| Other | NeuroTrace 435/455 blue-fluorescent Nissl stain | Invitrogen | Cat #: N21479 | IF (1:300) |
| Other | NeuroTrace 640/660 deep red-fluorescent Nissl stain | Invitrogen | RRID:AB_2572212 | IF (1:300) |
| Cell lines (*Mus musculus*) | Mouse cortical neuronal cells | Thermo Scientific | Cat#: A15585 | |
| Cell lines (*Mus musculus*) | Neuro2a (N2a) | ATCC | RRID:CVCL_0470 | |
| Cell lines (*Mus musculus*) | 4T1 | ATCC | RRID:CVCL_0125 | |
| Transfected construct (mouse) | Mouse Polk siRNA | Dharmacon | Cat#: A-048146-13-0050 | |

*Continued on next page*

*Continued*

| Reagent type (species) or resource | Designation | Source or reference | Identifiers | Additional information |
|---|---|---|---|---|
| Transfected construct (mouse) | Accell Red Non-targeting siRNA | Dharmacon | Cat#: D-001960-01-05 | |
| Transfected construct (mouse) | Mouse DinB siRNA | Santa Cruz | Cat#: sc-60538 | |
| Transfected construct (mouse) | Control siRNA-A | Santa Cruz | Cat#: sc-37007 | |

## Animals

Animal use approved and overseen by Penn State College of Medicine Institutional Animal Care and Committee and the Penn State College of Medicine Comparative Medicine PROTO201900781. The following mouse lines were used: wild-type C57/BL6 (Jackson Laboratory, Stock#000664), *Gad2$^{tm2(cre)}$$_{Zjh}$*/J (Jackson Laboratory, Stock#010802), and B6.Cg-*Gt(ROSA)26Sor$^{tm14CAG-tdTomato}$* (Jackson Laboratory, Stock #007914). Mice strains *Gad2$^{tm2(cre)Zjh}$*/J and B6.Cg-*Gt(ROSA)26Sor$^{tm14CAG-tdTomato}$* were bred together to generate Gad2; Ai14 for labeling and distinguishing interneurons for experiments under *Figures 4 and 5*. Mice were housed in a temperature and humidity-controlled environment in a barrier facility. Mice were kept under a standard 12 hr light–dark cycle, with food and water provided ad libitum. For experiments, animals were used at 1–3, 9–12, 18–19, and 24–27 months of age. No a priori sample size estimation was performed; group sizes were established on animal availability and experimental feasibility. Investigators were not blinded, as young and aged mice could be readily identified on the basis of overt age-related differences in physical appearances and behavior.

## Primary neuronal culture and siRNA transfection

Mouse cortical neuronal cells (Thermo Scientific, Cat No. A15585) were obtained from Thermo Scientific and prepared according to the manufacturer's protocol with minor modifications. Briefly, the frozen cells were rapidly thawed and resuspended in complete Neurobasal Medium (Cat No. 21103049, Gibco) supplemented with B-27 Supplement (Cat No. 17504044, Gibco) and GlutaMAX Supplement (Cat No. 35050061, Gibco). The cells were counted and seeded at a density of 100,000–500,000 cells/cm² on pre-coated 24-well plates (coated overnight with poly-D-lysine; Sigma-Aldrich, Cat No. P6407) or at a density of 40,000–200,000 cells/cm² on CC2 Glass 0.7 cm² 8-well chamber slides (Nunc Lab-Tek II CC2 Chamber Slide System, Thermo Scientific, Cat No. 154941). The plates and chamber slides were incubated in a 5% $CO_2$ incubator at 37°C. After 24 hr, the medium in each well was removed and replaced with fresh medium. The cells were fed every third day by aspirating half of the medium and replacing it with fresh medium.

At day in vitro 10 (DIV10), transfection of primary neuronal cultures with siRNA was performed according to the manufacturer's protocol (Horizon Dharmacon). Briefly, 1 µM of Accell siRNA was mixed with Accell siRNA Delivery Media (Cat No. B-005000; Horizon Discovery) and added to the neurons in complete Neurobasal Medium (50/50 vol/vol). The following siRNAs were used: Accell Mouse Polk siRNA (Cat No. A-048146-13-0050, Dharmacon), Accell Red Non-targeting siRNA (Cat No. D-001960-01-05, Dharmacon), or Accell Non-targeting siRNA pool (Cat No. D-001910-10-05, Dharmacon). A secondary method of transfection was performed using Lipofectamine RNAiMAX reagent (Thermo Fisher) and OptiMEM (Gibco) following the manufacturer's instructions. The siRNAs used were DinB siRNA (m) (sc-60538) and Control siRNA-A (Cat No. sc-37007). Cells were incubated with the siRNA complex in Opti-MEM for 6 hr, followed by replacement with complete Neurobasal medium.

## Immunocytochemistry

At 24, 48, and 72 hr after transfection, cortical primary neuronal cells were washed with DPBS+ (DPBS containing $Ca^{2+}$ and $Mg^{2+}$) and fixed with 4% PFA (Electron Microscopy Sciences, Cat No. 19208) for 20 min at room temperature. The cells were then washed with DPBS +three times. Permeabilization was achieved using 0.3% Triton X-100 (Thermo Scientific, Cat No. A16046 diluted in DPBS+) for 5 min at room temperature, followed by incubation in blocking solution (DPBS +containing 5% goat serum and 0.3% Triton X-100) for 1 hr. After washing, the cells were incubated with the respective primary antibodies (Key Resources Table) overnight. The next day, cells were incubated with

appropriate secondary antibodies (Key Resources Table) for 1 hr. Following additional washes, the chambers were separated from the glass slides, and cells were stained with Hoechst Nucleic Acid Stains (Cat No. H3570, Thermo Fisher Scientific). Coverslips were applied using mounting medium, and slides were stored at 4°C. Images were captured using a confocal microscope (LSM) and analyzed using NIH ImageJ software. All experiments were repeated at least three times, with details provided in the figure legends.

## IF assay

Mice were anesthetized with 100–200 µl of ketamine and underwent transcardiac perfusion using 1x phosphate-buffered saline (PBS, 10x diluted to 1x from Corning, Cat No. 46-013-CM) and 4% PFA made in 0.2 M PBS. Brain tissues were fixed with 4% PFA at 4°C for 24 hr, post-fixation brains were transferred to 1% sodium azide (Scientific inc, Cat No. DSS24080-250) in 1xPBS. Fixed brains were sectioned coronally at 60 µm thickness using a vibratome. For staining, free-floating brain slices underwent blocking consisting of 10% goat serum (Jackson ImmunoResearch, Cat No. 005000121), 1x PBS, and 10% Triton X-100 at RT for 2 hr on an orbital shaker. The following blocking slices were washed 3x with PBS at RT for 10 min each time. Slices were incubated with cocktail mix of 1x PBS, 1% Triton X-100, 5% goat serum, and primary antibody 4°C for 24 hr on an orbital shaker. Post-primary incubation, slices were washed 3x with PBS at RT for 10 min each time. Slices were incubated with 1xPBS, 1% Triton X-100, 5% goat serum, and secondary antibody at RT for 2 hr on an orbital shaker. Slices were washed for 10 min on orbital shaker at RT. Slices were incubated with 1xPBS and Nissl for 40 min at RT, followed by a final wash with 1xPBS for 10 min. Slices were mounted on ColorFrost Microscope slides using mounting media (Invitrogen, Cat No. E142757). IF assay in *Figure 1* used a slightly different procedure. The blocking buffer used 10% donkey serum instead of goat serum.

## Cell culture, transfection, and lentiviral transduction

Mouse Neuro2a (N2a) (CCL-131; ATCC, USA) and mouse 4T1 (CRL-2539; ATCC, USA) cell lines were obtained from and authenticated by ATCC, tested for mycoplasma using MycoAlert Mycoplasma Detection Kit (Lonza, Catalog #LT07-218), cultured in DMEM (Dulbecco's Modified Eagle Medium) and RPMI 1640 medium, respectively, both supplemented with 10% FBS, penicillin (100 IU/ml), and streptomycin (100 µg/ml) (Cat No. 30-002-CI, Corning) in a humidified incubator at 37°C with 5% $CO_2$. After 2 days, siRNA transfection was performed following the manufacturer's protocol (Horizon Dharmacon). Briefly, 1 µM of Accell siRNA was mixed with Accell siRNA Delivery Media and added to cells. The following siRNAs were used: Accell Mouse Polk siRNA (Cat No. A-048146-13-0050, Dharmacon, target sequence 5′ → 3′ CCAUUAAGCUGAAGAACGU and GCAUGGGACUAAACGA UAA) or Accell Non-targeting siRNA pool (Cat No. D-001910-10-05, Dharmacon, target sequence 5′ → 3′ UGGUUUACAUGUCGACUAA). An alternative transfection method was performed using Lipofectamine RNAiMAX reagent (Cat No. 13778150, Thermo Fisher) and OptiMEM (Gibco) following the manufacturer's instructions. The siRNAs used were DinB siRNA (mouse) (sc-60538, is a pool of three different siRNA duplexes sequences 5′ → 3′, sc-60538A: sense: GCACAGAACUCUACCAACAt t, antisense: UGUUGGUAGAGUUCUGUGCtt; sc-60538B: sense: CAAGAGACUUCUGAUCUUAtt, antisense: UAAGAUCAGAAGUCUCUUGtt; sc-60538C: sense: GGAAAUCAGUCGUGUUAGAtt, antisense: UCUAACACGACUGAUUUCCtt) and Accell Mouse Polk siRNA (Cat No. A-048146-13-0050, Dharmacon), along with Control siRNA-A (Cat No. sc-37007). Cells were incubated with the siRNA complex in Opti-MEM for 6 hr, followed by replacement with complete media. In addition, N2a cells were transduced with lentiviral vectors expressing shPolk (Cat No. TL502663V, Origene, which have four unique 29mer target-specific shRNAs sequences are 5′ → 3″, TL502663VA: AGCCATGCCAGG ATTTATTGCTAAGAGGC, TL502663VB: CCAGGATTTATTGCTAAGAGGCTCTGCC, TL502663VC: AATCGCAGCAAAGAGGAATGTCCTGATAT, TL502663VD: GGAGCTGCTAAGGACAGAAGTTAA TGTGG) or scrambled shRNA (Cat No. TR30021V, Origene, sequences are 5′ → 3″ GCACTACCAGAG CTAACTCAGATAGTACT) for 8 hr, followed by replacement with complete media. After 72 hr, cells were collected.

## Protein extraction and WB analysis

Total protein extracts from mouse brain cortex and primary neuronal cell cultures were prepared using the N-PER Neuronal Protein Extraction Reagent (Thermo Scientific, Cat No. 87792) with Halt

Protease Inhibitor Cocktail (Thermo Scientific, Cat No. 87786) following the manufacturer's instructions. Nuclear and cytoplasmic proteins from brain cortex tissues were extracted using the Minute Cytosolic and Nuclear Extraction Kit for Frozen/Fresh Tissues (Invent Biotechnologies INC, Cat No. NT-032), also following the manufacturer's instructions. Total protein extracts from N2A and 4T1 cells were prepared using RIPA buffer (Thermo Scientific, Cat No. 89900) with Halt Protease Inhibitor Cocktail. Protein concentration was determined using the Pierce BCA Protein Assay Kit (Thermo Scientific, Cat No. 23227). Protein samples were boiled at 95°C with 4x Laemmli sample buffer (Bio-Rad, Cat No. 1610747) for 5 min. Samples were separated on 4–20% Mini-PROTEAN TGX Precast Protein Gels (Bio-Rad) and transferred to a 0.45-µm LF PVDF membrane using the Trans-Blot Turbo Transfer System (Bio-Rad, Cat No. 1704274). After blocking with 5% nonfat dry milk in PBS with 0.1% Tween 20 for 1 hr at room temperature, membranes were incubated overnight at 4°C with primary antibodies: Polk (ABclonal), Polk-HRP (Santa Cruz Biotechnology), Alpha Tubulin, β-actin, GAPDH, and Histone-H3. Blots were further incubated with secondary antibody (HRP-conjugated anti-mouse IgG, 1:10,000) for 1 hr. Bands were detected using SuperSignal Chemiluminescent Substrate (Thermo Scientific) and scanned using the ChemiDoc Imaging System (Bio-Rad). Quantitative analyses were performed using ImageJ software (NIH). Proteins were normalized to the corresponding loading control.

## Confocal microscopy, image analysis, quantification

Imaging was performed using Zeiss LSM910 with AiryScan2 super-resolution in multiplex mode SR-2Y, using 40X water (NA = 1.2) and 63X oil (NA = 1.4) objectives. All images were captured at 16-bit depth. For the 40X objective, scaling per pixel is 0.078 µm × 0.078 µm with an image size of 4084 × 4084 pixels. The scaling for the 63X objective is 0.035 µm x 0.035 µm per pixel with an image size of 2860 × 2860 pixels. The imaging parameters such as laser power, digital gain, digital offset, and scan-speed were kept constants for all samples within each experiment. Images were saved as uncompressed .czi files, and imported into the CellProfiler v4.2.6 software, where cells were registered, subcellularly segmented, and POLK nuclear speckles and cytoplasmic granules were detected as individual objects and quantified.

## Statistical analysis

Statistical analysis, Wilcoxon rank-sum, ANOVA, ANCOVA, Random Forest, and Boosting Classifier were done using open-source software JASP 0.18.3, R Studio v2023.06.1+524, and GraphPad Prism. Plots were generated using R Studio, Python, JASP, and GraphPad, exported and composed in Adobe Illustrator.

## Ex vivo KA treatment

Brains from 1- and 18-month-old C57BL/6J animals were dissected and immediately placed on ice. Brains were placed on the mount and cut through the midline, separating each hemisphere. 1.5% agar was prepared and used for the mold of the brain. The brains were submerged in the ice-cold, oxygenated sucrose dissecting solution (in mM: 183 sucrose, 20 NaCl, 0.5 KCl, 1 MgCl$_2$, 1.4 NaH$_2$PO$_4$, 25 NaHCO$_3$, and 10 glucose). Coronal hemi-slices of 100 µm thickness for IF experiments were prepared using a Compresstome VF-310-0Z (Precisionary Instruments, LLC) with a double-edged stainless-steel blade. Both hemi-slices were placed in an oxygenated holding solution with a modified ACSF (in mM: 100 sucrose, 60 NaCl, 2.5 KCl, 1.4 NaH$_2$PO$_4$, 1.1 CaCl$_2$, 3.2 MgCl$_2$, 1.2 MgSO$_4$, 22 NaHCO$_3$, 20 glucose, 1 ascorbic acid) for 14 min. After 14 min, the hemi-slices for control were transferred to an oxygenated standard ACSF solution (in mM: 124 NaCl, 4.4 KCl, 2 CaCl$_2$, 2.95 MgSO$_4$, 1 NaH$_2$PO$_4$, 10 glucose, 26 NaHCO$_3$) at 28–32°C. The treatment group received bath application of 1.0 µM KA (Millipore Sigma, Cat No. K0250) dissolved in ACSF. Control and KA treated slices were removed from the bath at 80 and 160 min. For IF experiments, slices were fixed in 4% PFA for 30 min followed by storage in 1% sodium azide to proceed with IF protocol.

## Figure generation

All representative images from the Zeiss.czi files were exported as TIFFs or JPEGs using Zen, FIJI, and CellProfiler. Graphical plots were exported as PDFs from RStudio, JASP, or GraphPad Prism, and subpanels were assembled into final figures using Adobe Illustrator.

## Acknowledgements

Work was supported by grant NIH RF1AG072602/R01AG072602 to AP and Startup Funds from Penn State College of Medicine to AP.

## Additional information

### Funding

| Funder | Grant reference number | Author |
| --- | --- | --- |
| National Institute on Aging | NIH RF1AG072602 / R01AG072602 | Anirban Paul |
| Pennsylvania State University | Startup funds | Anirban Paul |

The funders had no role in study design, data collection, and interpretation, or the decision to submit the work for publication.

### Author contributions

Mofida Abdelmageed, Premkumar Palanisamy, Data curation, Formal analysis, Validation, Investigation, Visualization, Methodology, Writing – review and editing; Victoria Vernail, Yuval Silberman, Methodology; Shilpi Paul, Conceptualization, Resources, Formal analysis, Supervision, Visualization, Writing – original draft, Project administration, Writing – review and editing; Anirban Paul, Conceptualization, Resources, Software, Formal analysis, Supervision, Funding acquisition, Visualization, Methodology, Writing – original draft, Project administration, Writing – review and editing

### Author ORCIDs

Mofida Abdelmageed ⬥ https://orcid.org/0000-0001-9507-4296
Premkumar Palanisamy ⬥ https://orcid.org/0000-0002-8255-0625
Victoria Vernail ⬥ https://orcid.org/0000-0002-4910-9280
Yuval Silberman ⬥ https://orcid.org/0000-0002-4694-4053
Shilpi Paul ⬥ https://orcid.org/0000-0001-5075-2081
Anirban Paul ⬥ https://orcid.org/0000-0001-5347-9260

### Ethics

This study was performed in strict accordance with the recommendations in the Guide for the Care and Use of Laboratory Animals of the National Institutes of Health. All of the animals were handled according to approved Institutional Animal Care and Use Committee (IACUC) protocol #PROTO202101837 of Penn State University College of Medicine.

Reviewer #1 (Public review): https://doi.org/10.7554/eLife.101533.3.sa1
Reviewer #2 (Public review): https://doi.org/10.7554/eLife.101533.3.sa2
Author response https://doi.org/10.7554/eLife.101533.3.sa3

## Additional files

### Supplementary files

MDAR checklist

### Data availability

This study did not generate any new RNA sequencing or proteomics datasets. All data supporting the findings of this study are provided within the main text and source data files.

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
