## [Editor Report · eLife Assessment]

This manuscript details **important** findings that DNA polymerase kappa shows age-related changes in subcellular localization within different cell types in the brains of mice, from the nucleus in young cells to the cytoplasm in old cells. The authors' findings suggest that age-related alterations in POLK localization could drive mechanistic and functional changes in the aging brain. The authors provide **solid** evidence for their study, with data broadly supporting their claims with minor weaknesses.

---

## [Referee Report · Reviewer #1 (Public review)]

Summary:

Abdelmageed et al. investigate age-related changes in the subcellular localization of DNA polymerase kappa (POLK) in the brains of mice. POLK has been actively investigated for its role in translesion DNA synthesis and involvement in other DNA repair pathways in proliferating cells, very little is known about POLK in a tissue-specific context or let alone in post-mitotic cells. The authors investigated POLK subcellular distribution in the brains of young, middle-aged, and old mice via immunoblotting of fractioned tissue extracts and immunofluorescence (IF). Immunoblotting revealed a progressive decrease in the abundance of nuclear POLK, while cytoplasmic POLK levels concomitantly increased. Similar findings were present when IF was performed on brain sections. Further IF studies of cingulate cortex (Cg1), motor cortex (M1, M2), and somatosensory (S1) cortical regions all showed an age-related decline in nuclear POLK. Nuclear speckles of POLK decrease in each region, meanwhile the number of cytoplasmic POLK granules decreases in all four regions, but granule size is increasing. The authors report similar findings for REV1, another Y-family DNA polymerase.

The authors then investigate the colocalization of POLK with other DNA damage response (DDR) proteins in either pyramidal neurons or inhibitory interneurons. At 18 months of age, DNA damage marker gH2AX demonstrated colocalization with nuclear POLK, while strong colocalization of POLK and 8-oxo-dG was present in geriatric mice. The authors find that cytoplasmic POLK granules colocalize with stress granule marker G3BP1, suggesting that the accumulated POLK ends up in the lysosome.

Brain regions were further stained to identify POLK patterns in NeuN+ neurons, GABAergic neurons, and other non-neuronal cell types present in the cortex. Microglia associated with pyramidal neurons or inhibitory interneurons were found to have higher abundance of cytoplasmic POLK. The authors also report that POLK localization can be regulated by neuronal activity induced by Kainic acid treatment. Lastly, the authors suggest that POLK could serve as an aging clock for brain tissue, but POLK deserves further characterization and correlation to functional changes before being considered for a biomarker.

Strengths:

Investigation of TLS polymerases in specific tissues and in post-mitotic cells is largely understudied. The potential changes in sub cellular localization of POLK and potentially other TLS polymerases opens up many questions about DNA repair and damage tolerance in the brain and how it can change with age.

Weaknesses:

The work is quite novel and interesting, and the authors do suggest some potentially interesting roles for POLK in the brain, but these are in of themselves a bit speculative. The majority of the findings of this paper draw upon findings from POLK antibody and its presumed specificity for POLK. However, this antibody has not been fully validated and would benefit from further validation of the different band sizes. More mechanistic investigation is needed before POLK could be considered as a brain aging clock but does not preclude the potential for using POLK as a biological "dating" system for the brain.

Comments on revisions:

The revised manuscript is suitably improved and addresses reviewer comments.

---

## [Referee Report · Reviewer #2 (Public review)]

Summary:

Abdelmageed et al., demonstrate POLK expression in nervous tissue and focus mainly on neurons. Here, they describe an exciting age-dependent change in POLK subcellular localization, from the nucleus in young tissue to the cytoplasm in old tissue. They argue that the cytosolic POLK associates with stress granules. They also investigate cell-type specific expression of POLK, and quantitate expression changes induced by cell autonomous (activity) and cell nonautonomous (microglia) factors.

Comments on revisions:

Do the authors have any explanation or reason for why they weren't able to achieve a higher knockdown of POLK using siRNA in Figure 1A2? It does not seem statistically different by eye, as all values in the KD overlap with the control. This does not seem like strong evidence that their antibody works.

---

## [Author Response]

The following is the authors’ response to the original reviews.

**Public Reviews:**

**Reviewer #1 (Public review):**
Summary:Abdelmageed et al. investigate age-related changes in the subcellular localization of DNA polymerase kappa (POLK) in the brains of mice. POLK has been actively investigated for its role in translesion DNA synthesis and involvement in other DNA repair pathways in proliferating cells, very little is known about POLK in a tissue-specific context, let alone in post-mitotic cells. The authors investigated POLK subcellular distribution in the brains of young, middle-aged, and old mice via immunoblotting of fractioned tissue extracts and immunofluorescence (IF). Immunoblotting revealed a progressive decrease in the abundance of nuclear POLK, while cytoplasmic POLK levels concomitantly increased. Similar findings were present when IF was performed on brain sections. Further, IF studies of the cingulate cortex (Cg1), the motor cortex (M1, M2), and the somatosensory (S1) cortical regions all showed an age-related decline in nuclear POLK. Nuclear speckles of POLK decrease in each region, meanwhile, the number of cytoplasmic POLK granules decreases in all four regions, but granule size is increasing. The authors report similar findings for REV1, another Y-family DNA polymerase.The authors then investigate the colocalization of POLK with other DNA damage response (DDR) proteins in either pyramidal neurons or inhibitory interneurons. At 18 months of age, DNA damage marker gH2AX demonstrated colocalization with nuclear POLK, while strong colocalization of POLK and 8-oxo-dG was present in geriatric mice. The authors find that cytoplasmic POLK granules colocalize with stress granule marker G3BP1, suggesting that the accumulated POLK ends up in the lysosome.Brain regions were further stained to identify POLK patterns in NeuN+ neurons, GABAergic neurons, and other non-neuronal cell types present in the cortex. Microglia associated with pyramidal neurons or inhibitory interneurons were found to have a higher abundance of cytoplasmic POLK. The authors also report that POLK localization can be regulated by neuronal activity induced by Kainic acid treatment. Lastly, the authors suggest that POLK could serve as an aging clock for brain tissue, but POLK deserves further characterization and correlation to functional changes before being considered as a biomarker.Strengths:Investigation of TLS polymerases in specific tissues and in post-mitotic cells is largely understudied. The potential changes in sub-cellular localization of POLK and potentially other TLS polymerases open up many questions about DNA repair and damage tolerance in the brain and how it can change with age.Weaknesses:The work is quite novel and interesting, and the authors do suggest some potentially interesting roles for POLK in the brain, but these are in and of themselves a bit speculative. The majority of the findings of this paper draw upon findings from POLK antibody and its presumed specificity for POLK. However, this antibody has not been fully validated and needs further work. Further validation experiments using Polk-deficient or knocked-down cells to investigate antibody specificity for both immunoblotting and immunofluorescence should be performed. More mechanistic investigation is needed before POLK could be considered as a brain aging clock.

We are thankful for the overall enthusiasm and positive comments.

(a) Concern over POLK antibody characterization in mouse:

We performed siRNA and shRNA knock downs in mouse primary cortical neurons as well as efficiently transfectable murine lines like 4T1 and Neuro-2A showing knock down of 99kDa and 120kDa bands recognized by sc-166667 anti-POLK antibody (exact figure number Figure 1 and S1). We show that in IF sc-166667 and A12052 (Figure S1G) shows similar immunostaining patterns and we used sc-166667 in all reported figures and western blots.

(b) More mechanistic investigation is needed before POLK could be considered as a brain aging clock:

We sincerely appreciate the valuable suggestion. We agree as a terminal assay POLK nucleo-cytoplasmic status is not practical for longitudinal studies. However, we believe it may serve an investigative/correlative endogenous signal for determining tissue age, that may be useful to "date" brain sections, since not many such cell biological markers exist. We have added clarification texts to address this.

**Reviewer #2 (Public review):**
Summary:Abdelmageed et al., demonstrate POLK expression in nervous tissue and focus mainly on neurons. Here they describe an exciting age-dependent change in POLK subcellular localization, from the nucleus in young tissue to the cytoplasm in old tissue. They argue that the cytosolic POLK is associated with stress granules. They also investigate the cell-type specific expression of POLK, and quantitate expression changes induced by cell-autonomous (activity) and cell nonautonomous (microglia) factors.I think it is an interesting report but requires a few more experiments to support their findings in the latter half of the paper. Additionally, a more mechanistic understanding of the pathways regulating POLK dynamics between the nucleus and cytosol, what is POLK doing in the cytosol, and what is it interacting with; would greatly increase the impact of this report. However, additional mechanistic experiments are mostly not needed to support much of the currently presented results, again, it would simply increase the impact.

(a) Concern on more mechanistic understanding of the pathways regulating POLK dynamics between the nucleus and cytosol:

We sincerely appreciate the reviewer’s enthusiasm and valuable guidance in helping us better understand the mechanism of nuclear-cytoplasmic POLK dynamics. Previously, we developed a modified aniPOND (accelerated native isolation of proteins on nascent DNA) protocol, which we termed iPoKD-MS (isolation of proteins on Pol kappa synthesized DNA followed by mass spectrometry), to capture proteins bound to nascent DNA synthesized by POLK in human cell lines (bioRxiv https://www.biorxiv.org/content/10.1101/2022.10.27.513845v3). In this dataset, we identified potential candidates that may regulate nuclear/cytoplasmic POLK dynamics. These candidates are currently undergoing validation in human cell lines, and we are preparing a manuscript on these findings. Among these, some candidates, including previously identified proteins such as exportin and importin (Temprine et al., 2020, PMID: 32345725), are being explored further as potential POLK nuclear/cytoplasmic shuttles. We are also conducting tests on these candidates in mouse cortical primary neurons to assess their role in POLK dynamics. In the revised version of the manuscript, we have included a discussion of our current understanding.

(b) Question on “… what is POLK doing in the cytosol, and what is it interacting with …”: Our data so far indicate that POLK accumulates in stress granules and lysosomes. We are very grateful for the reviewer’s insightful suggestions and will make every effort to incorporate them in the revised manuscript. We characterized POLK accumulation in the cytoplasm using six additional endo-lysosomal markers, as recommended by the reviewer. This data is now part of entirely new Figure 3.

**Reviewer #3 (Public review):**
Summary:In this study, the authors show that DNA polymerase kappa POLK relocalizes in the cytoplasm as granules with age in mice. The reduction of nuclear POLK in old brains is congruent with an increase in DNA damage markers. The cytoplasmic granules colocalize with stress granules and endo-lysosome. The study proposes that protein localization of POLK could be used to determine the biological age of brain tissue sections.Strengths:Very few studies focus on the POLK protein in the peripheral nervous system (PNS). The microscopy approach used here is also very relevant: it allows the authors to highlight a radical change in POLK localization (nuclear versus cytoplasmic) depending on the age of the neurons.The conclusions of the study are strong. Several types of neurons are compared, the colocalization with several proteins from the NHEJ and BER repair pathways is tested, and microscopy images are systematically quantified.Weaknesses:The authors do not discuss the physical nature of POLK granules. There is a large field of research dedicated to the nature and function of condensates: in particular numerous studies have shown that some condensates but not all exhibit liquid-like properties (https://www.nature.com/articles/nrm.2017.7, https://pubmed.ncbi.nlm.nih.gov/33510441/
https://www.mdpi.com/2073-4425/13/10/1846). The change of physical properties of condensates is particularly important in cells undergoing stress and during aging. The authors should discuss this literature.

We highly appreciate the reviewer bringing up the context of biomolecular condensates. Our iPoKD-MS data referenced above suggests candidates from various biomolecular condensates that we are currently investigating. We appreciate the reviewer providing important literature cited these articles in text and potential biomolecular condensates are discussed in the revised version.

**Recommendations for the authors:**

**Reviewer #1 (Recommendations for the authors):**
The work is quite novel and interesting, and the authors do suggest some potentially interesting roles for POLK in the brain, but these are in of themselves a bit speculative. The majority of the findings of this paper rely upon the POLK antibody and its specificity for POLK, which is not fully characterized and needs further work (validation of antibodies using immunoblots of Polk KO cells or siRNA KD of POLK in murine cells) to provide confidence in the authors' findings.PointssiRNA knockdown of Polk in primary neurons showed a dramatic reduction in signal by IF even though qPCR analysis showed a reduction of only ~35% at the transcript level. Typically many DNA repair genes need to be knocked down by 80% or more to see discernable differences at the protein level. siRNA knockdown in a murine cell line (MEFs, neurons, or some other easily transfectable cell type) needs to be performed with immunoblotting with whole cell and fractionated (nuclear/cytoplasmic) lysates in order to better validate the anti-POLK antibodies and which bands that are visualized during immunoblotting are specific to POLK.

We performed siRNA and shRNA knock downs in mouse primary cortical neurons as well as efficiently transfectable murine lines like 4T1 and Neuro-2A showing knock down of 99kDa and 120kDa bands recognized by sc-166667 anti-POLK antibody (exact figure number Figure 1 and S1). We show that in IF sc-166667 and A12052 (Figure S1G) shows similar immunostaining patterns and we used sc-166667 in all reported figures and western blots.

Figure 1B and C, it is not clear which antibody(ies) are used for the immunoblotting of nuclear and cytoplasmic fractions and for a blot with whole tissue lysates. Please place the antibody vendor or clone next to the corresponding blot or describe it in the figure legend. Bands of varying sizes are present in 1B (and Figure S1) but only a band at 99 kDa was shown in 1C. Because there are no bands of equivalent size present in the nuclear and cytoplasmic fractions in Figure 1B, please describe or denote which bands were used for quantification purposes for nuclear and cytoplasmic POLK.

This has been clarified by using only one antibody throughout the manuscript sc-166667. We observed in whole cell lysate an intense ~99kDa and a faint ~120kDa band, which gets intense in nuclear fraction and is absent in cytoplasmic fraction. We have noted this in multiple human cell lines and hiPSC-derived neurons, which is our ongoing work. We do not know yet if the ~120kDa is a modification or isoform of POLK. We have hints from our proteomics data that it may be SUMOylated or ubiquitinylated or other post translational modifications. We added this in the discussion section.

Figure 1I, is there a quantification beyond just the representative image? There is no green staining pattern outside the cytoplasm in the 1-month-old M1 images that is present in all the other images in the panel.

Fig 1I is now Fig S1G in the revised manuscript. Since REV1 and POLH were not central to the study that focused on POLK, they were meant to be exploratory data panels and as such we did not quantify beyond the qualitative evaluation, which broadly resembled POLK’s disposition with age. We have noted there are some sample to sample variability in the background signal. In general, outside the cytoplasm as subcellularly segmented by fluorescent nissl expression, tends to be variable by brain areas but also higher in older brains

"Association with PRKDC further suggests POLK's role in the "gap-filling" step in the NHEJ repair pathway in neurons." There is no strong evidence in the literature for mammalian POLK playing a role in NHEJ. Some description of a role in HR has been described, however. The reference regarding the iPoKD-MS data set that provides evidence of POLK associating with BER and NHEJ factors is listed as Paul, 2022 but is in the reference list as Shilpi Paul 2022.

We removed this speculative statement and citation fixed.

Figure 4A, what is the age of the mouse for the representative images?

19 months and now mentioned in the figure legend

Figure 4C, Could the data from the different ages be plotted side by side to better evaluate the differences for each cell type/region?

Data is plotted side by side

Why was the one-month time point chosen as this could still represent the developing and not mature murine brain?

Reviewer correctly noted that a 1 month brain is still developing, but mostly from the behavioral and circuit maturation standpoint. However, from cell division and neurogenesis perspective, that is considered to be complete by first postnatal month, with neuron production thereafter largely restricted to specialized adult niches in the dentate gyrus and subventricular zone–olfactory bulb pathway; these adult neurogenic stem cells are embryonically derived and are regulated in ways that are distinct from the early, expansionary developmental waves of neurogenesis. In our study we performed our measurements in the cortical areas only. (Caviness et al., 1995, PMID: 7482802; Ansorg et al., 2012, PMID: 22564330; Ming & Song, 2011, PMID: 21609825; Bond et al., 2015, PMID: 26431181; Bond et al., 2021, PMID: 33706926; Bartkowska et al., 2022, PMID: 36078144). Also, in Figure 6A it was incorrectly mentioned to be just 1month, we rechecked our metadata and noted that young brains were comprised of 1 and 2 month old brains and now it has been corrected.

Furthermore, can the authors describe which sex of mice was used in these experiments and the justification if a single sex was used? If both sexes were used, were there any dimorphic differences in POLK localization patterns?

This is an important aspect, but in the beginning to keep mice numbers within manageable limits, we were focusing more on the age component. While both males and female brains were assayed but due to uneven sample distribution between sexes, we could not estimate if there were any statistically significant sexual dimorphic differences in IN, PN and NNs. Future studies will investigate the sex component as a function of age.

The suggestion of POLK as a brain aging clock may be a bit premature as the functional and behavioral consequences of cytoplasmic POLK sequestration are not fully known. Furthermore, investigation of POLK levels in other genetic models of neurodegeneration or with gerotherapeutics would be needed to establish if the POLK brain clock is responsive to changes that shift brain aging. Lastly, this clock may be impractical and not useful for longitudinal studies due to the terminal nature of assessing POLK levels.

We agree as a terminal assay POLK nucleo-cytoplasmic status is not practical for longitudinal studies. However, we believe it may serve an investigative/correlative endogenous signal for determining tissue age, that may be useful to "date" brain sections, since not many such cell biological markers exist. We have added clarification text.

Some discussion of the Polk-null mice is warranted, as they only have a slightly shortened lifespan, and any disease phenotypes were not reported. This stands in contrast to other DNA repair-deficient mice that mimic premature aging and show behavioral and motor deficits. This calls into question the role of POLK in brain aging.

Discussion statements on Polk-null mice has been added.

Please correct the catalog number for the SCBT anti-POLK antibody to sc-166667

Typographical error has been corrected

**Reviewer #2 (Recommendations for the authors):**
Results:Figure by figure(1) A progressive age-associated shift in subcellular localization of POLK The authors state that POLK has not been studied in nervous tissue before and they want to see if it is expressed, and if it changes subcellular location as a function of age. The authors argue age = stress like that seen in previous models using genotoxic agents and cancer cells. Indeed, POLK seems to convincingly change subcellular location from the nucleus to larger cytosolic puncta.(2) Nuclear POLK co-localizes with DNA damage response and repair proteins This was a difficult dataset for me to decipher. To me, it appears as though POLK colocalizes with these examined proteins in the CYTOSOL, not the nucleus. Especially, in the oldest mice.

We added in the discussion that DNA repair proteins were observed to be present in the cytoplasm and biomolecular condensates citing relevant reviews and primary references.

(3) POLK in the cytoplasm is associated with stress granules and lysosomes in old brains LAMP1 has some issues as a lysosome marker. The authors even state it can be on endosomes. It would be nice to use a marker for mature lysosomes, some fluorescent reporter that is activated only by lysosomal proteases or pH. It is also of interest if POLK is localized to the membrane or the inside of these structures. The authors have access to an airyscan which is sufficient to examine luminal vs membrane localization on larger organelles like lysosomes.

We thank the reviewer for pushing us to investigate the nature of cytoplasmic POLK in endo-lysosomal compartments. We have now added a full-page figure on the cell biological results from six different markers, subset (Cathepsin B and D) are known to present in the lumens of endo-lysosomes, in Figure 3. Further high-resolution membrane vs lumen was not pursued, which is perhaps better suited in cultured neurons rather than thick fixed tissues.

(4) Differentially altered POLK subcellular expression amongst excitatory, inhibitory, and nonneuronal cells in the cortex.This seems fine. I don't see anything wrong with the author's statement that there is more POLK in neurons vs non-neuronal cells.(5) Microglia associated with IN and PN have significantly higher levels of cytoplasmic POLK I don't see really any convincing evidence of the author's claim here. They find a difference at early-old age, but not at old-old, or other ages. This is explained by "However, this effect is lost in late-old age (Figure 5D), likely due to the MG-mediated removal of the INs.". But no trend being observed, no experiment to show sufficiency, and no experiment to uncover a directional relationship; this is a tough claim to stand by.

Changes made in text to reflect speculative nature of this observation

(6) Subcellular localization of POLK is regulated by neuronal activityInteresting and fairly difficult experiment. Can the authors talk more about what these values mean? I am confused as to why there is a decline in nuclear puncta at 80 min. Also, why are POLK counts in 6c similar at baseline between young and early-old? In Figures 5 and 6 I also worry about statistical analysis. Are all assumptions checked to use t-tests? Why not always use a test that has fewer assumptions?

We have explained in the text the artificial nature of few hour long acute slice preparations is very different and inherently a stressful environment, especially for the old brains, compared to the vascular perfused PFA fixed brain tissues tested between young and old ages.

We don’t have a proper explanation for the initial dip in nuclear puncta in both young and old brains at 80min of very similar magnitude. It could be a separate biological phenomenon that occurs at much shorter time scales that would not otherwise be captured in a fixed tissue assay and needs careful investigation using live tissue fluorescence imaging that is beyond the scope of this manuscript.

We apologize for the typographical error in the figure legend. We rechecked our R code and the tests were all Wilcoxon rank-sum (Mann–Whitney U) two-sided nonparametric.

Figure 6B & E had absurdly small p values due to large sample numbers. So, we implemented random sampling of 100 cells repeating for 200 times and presented the distribution of p values and Cohen’s d in the supplement and reported the median p value and Cohen’s in the main plot.

(7) POLK as an endogenous "aging clock" for brain tissueTrainable model. What are the criteria for the model, and how does it work? The cutoffs it uses to classify each age group might be interesting in that the model may have identified a trait the researchers were unaware of. Otherwise, it is not especially useful. Maybe as an independent 'blind' analysis of the data?

We have added a better description of the models, assumptions and how two different unsupervised approaches converge on the same set of features with high AUROCs.

Minor questions:The cartoons (1a, 2a-b, 5a, 6a) help a lot. However, I still had to work a bit to understand some of the graphs (e.g., 5d, 6b-e, fig 7). Is there a simpler way to present them? Maybe simply additional labelling? I'm not sure.A more thorough discussion of statistical tests is warranted I think. I am not very clear why some were chosen (t-test vs nonparametric with fewer assumptions). Infinitesimally small p values also make me think maybe incorrect tests were done or no power analysis was performed beforehand. A fix for this is just discussing what went into the testing methods and why they were chosen.

Statistical analysis for Fig2 (using Generalized Estimating Equations), and Fig6 (with random repeated subsampling; method explained in text, figure legend updated and supplementary data on the distribution of p values and cohen’s d are added) to address the very small p values. Descriptions rewritten in relevant text.

In the absence of further mechanistic experiments, it would still be interesting to hear what the authors think is going on and what the significance of this altered subcellular location means. How do the authors think this is occurring? I think they are arguing that cytosolic localization of POLK is 100% detrimental to the neuron. ("The reduction of nuclear POLK in old brains is congruent with an increase in DNA damage markers") Do they have any idea what the 'bug' is in the POLK system then?

Statements in the discussion has been added.

**Reviewer #3 (Recommendations for the authors):**
POLK is detected as small " as small "speckles" inside the nucleus at a young age (1-2 months) and larger "granules" can be seen in the cytoplasm at progressively older time points (>9 months). In the nucleus, is POLK bound to DNA? In the cytoplasm, how are the POLK molecules organized: are they bound to a substrate or are they just organized as a proteins condensate without DNA?

In human U2OS cell line Dnase1 treatment leads to loss of POLK from the nucleus as well as its activity as reported in Fig5 of Paul, S. et. al. 2023 bioRxiv. While we haven’t reproduced these results in mouse primary neurons, we anticipate a similar situation which will be tested in the future. We have addressed limited aspects of the POLK in the cytoplasm in all new Fig3 with six endo-lysosomal markers, and added text.

When POLK proteins accumulate in the cytoplasm in aging cells, do they also repair condensates in the cytoplasm? What is the function of cytoplasmic POLK granules? More generally, is it known if other granules or foci, such as repair foci are found in the cytoplasms in aging cells, or in cells under stress?

Six markers for endo-lysosomes were tested to characterize the cytoplasmic granules now shown in Fig3.

While the authors quantify the number and sizes of the POLK signal, they don't discuss their physical nature. Some membrane-less condensates exhibit liquid-like properties, such as stress granules, P-bodies, or in the nucleus some repair condensates. In some diseased tissues, some condensates lose their liquid properties and become solid-like. Is it known if POLK condensates behave like liquid condensates or they are simply formed by bound molecules on DNA? Since they are larger and fewer in the cytoplasm, is it because several small puncta fused together to form a larger one? It would be worthwhile to discuss these points.

Discussion statements on the nature of condensates in context of the POLK cytoplasmic signal has been added.